# Horizontal transmission maintains host specificity and codiversification of symbionts in a brood parasitic host

Luiz Gustavo A. Pedroso [1,2,3✉], Pavel B. Klimov [2,4,5✉], Sergey V. Mironov [6], Barry M. OConnor[3], Henk R. Braig[5,7], Almir R. Pepato[8], Kevin P. Johnson [9], Qixin He [2✉] & Fabio Akashi Hernandes[1,10]

In host-symbiont systems, interspecific transmissions create opportunities for host switches, potentially leading to cophylogenetic incongruence. In contrast, conspecific transmissions often result in high host specificity and congruent cophylogenies. In most bird-feather mite systems, conspecific transmission is considered dominant, while interspecific transmission is supposedly rare. However, while mites typically maintain high host specificity, incongruent cophylogenies are common. To explain this conundrum, we quantify the magnitude of conspecific vs. interspecific transmission in the brood parasitic shiny cowbird (*Molothrus bonariensis*). *M. bonariensis* lacks parental care, allowing the assessment of the role of horizontal transmission alone in maintaining host specificity. We found that despite frequent interspecific interactions via foster parental care, mite species dispersing via conspecific horizontal contacts are three times more likely to colonize *M. bonariensis* than mites transmitted vertically via foster parents. The results highlight the previously underappreciated rate of transmission via horizontal contacts in maintaining host specificity on a microevolutionary scale. On a macroevolutionary scale, however, host switches were estimated to have occurred as frequently as codivergences. This suggests that macroevolutionary patterns resulting from rare events cannot be easily generalized from short-term evolutionary trends.

[1] Departamento de Zoologia, Av. 24-A, 1515, 13506-900, Universidade Estadual Paulista, Rio Claro, São Paulo State, Brazil. [2] Department of Biological Sciences, Purdue University, West Lafayette, IN, USA. [3] Department of Ecology and Evolutionary Biology, Museum of Zoology, University of Michigan, Ann Arbor, MI, USA. [4] Tyumen State University, 10 Semakova Str., 625003 Tyumen, Russia. [5] Bangor University, Brambell 503, School of Natural Sciences, Bangor LL57 2 UW Wales, UK. [6] Zoological Institute of the Russian Academy of Sciences, Saint Petersburg 199034, Russia. [7] Institute and Museum of Natural Sciences, Faculty of Natural and Exact Sciences, National University of San Juan, San Juan, Argentina. [8] Departamento de Zoologia, Instituto de Ciências Biológicas, Universidade Federal de Minas Gerais, Belo Horizonte, Brazil. [9] Illinois Natural History Survey, Prairie Research Institute, University of Illinois Urbana-Champaign, Champaign, IL, USA. [10] Departamento de Ecologia e Zoologia, CCB/ECZ, Trindade, Universidade Federal de Santa Catarina, 88040-970 Florianópolis, Santa Catarina, Brazil. ✉email: luizgustavopedroso@gmail.com; pklimov@purdue.edu; heqixin@purdue.edu

With the advance of cophylogenetic analytical and methodological frameworks, many host-symbiont systems have been assessed for the relative contribution of codiversification versus other types of events that shape coevolutionary histories of hosts and symbionts. The majority of studies have suggested that strict codiversification between hosts and symbionts (i.e., temporal and topological congruence of host and parasite phylogenetic branching pattern) is rare[1–14], on average, being only 7% as common as other coevolutionary events[15]. Generally, cophylogenetic incongruence (i.e., the disagreement between host and symbiont phylogenetic branching patterns at the macroevolutionary scale) may be caused by several evolutionary events, such as duplication (speciation of a symbiont within a single-host species), sorting (extinction and missing the boat), failure of the symbiont to speciate, and host switching (or host shift)[16]. Among these events, host switching is typically the most frequent event[1,4,9,16,17]. At the microevolutionary scale, host switching is also a biologically intriguing event leading to the evolution of multihost symbionts, especially when it occurs between phylogenetically distant hosts[1,18,19]. Most host switches occur via interspecific horizontal transfers (Fig. 1: $q_{hi}$), promoting incongruence in host and symbiont phylogenies[20–22]. In contrast, conspecific vertical transmission, i.e., from biological parents to offspring (Fig. 1: $q_{vc}$), is expected to maintain single-host symbionts (i.e., high host specificity) and produce congruent host and symbiont phylogenies (strict codiversification)[22,23]. Yet, despite the perceived dominance of vertical transmission and low horizontal transmission rates[24–28], some host-symbiont systems may simultaneously display both incongruent cophylogenetic patterns and high host specificity[3,5,16,29,30]. This conundrum challenges the role of vertical conspecific transmission in promoting codiversification and maintaining host specificity.

A major and well-studied host-symbiont system is that of birds and feather mites (Acariformes: Analgoidea and Pterolichoidea), where mites have high levels of dependence and specificity with their avian hosts[31–34]. These symbiotic organisms spend their entire life cycle on their host (full-time, obligate symbionts). With a few exceptions (e.g., some skin mites), they do not have a specialized dispersal stage and seem to lack any other adaptations for long-range dispersal between hosts[33,35,36]. Therefore, the most important dispersal mode of feather mites across host individuals should be via parental care, i.e., vertical conspecific transmission from host parents to offspring (Fig. 1A: $q_{vc}$)[16,25,37–39]. Consequently, the diversification of these symbionts is expected to be driven largely by host evolution. However, multiple cophylogenetic studies have shown that host switches are relatively common in feather mites[3,5,6,32,40], suggesting that host switches are in fact one of the main drivers of feather mite diversification[3]. Thus, the current biological expectations are in conflict: one suggests that vertical transmission should be prevalent, leading to congruence between host and symbiont phylogenies, while observations show widespread phylogenetic incongruence among mites, despite their high host specificity. Therefore, it is crucial to understand the relative contribution of two types of conspecific transmission: vertical and horizontal ($q_{vc}$ and $q_{hc}$), promoting cophylogenetic concordance and high host specificity vs interspecific transmission ($q_{vi}$ and $q_{hi}$), that can generate cophylogenetic discordance and low host specificity (Fig. 1). In feather mites, vertical conspecific transmission ($q_{vc}$) occurring from parents to chicks during the nesting period can be accurately measured[25,39]. However, quantifying conspecific horizontal transmission ($q_{hc}$), which occurs in the form of social transmission (physical contact between hosts), is particularly difficult as it takes place outside of the nesting period. As a result, vertical transmission ($q_{vc}$) has been overemphasized while conspecific horizontal transmission ($q_{hc}$) has been largely overlooked in the literature. This has contributed to the uncertainty regarding the role of conspecific horizontal transmission in shaping both host specificity and cophylogenetic congruence.

The brood parasitic shiny cowbird, *Molothrus bonariensis* (Passeriformes: Icteridae), provides an excellent model to investigate the relationship between conspecific and interspecific transmission and host specificity in feather mites. Like all obligate brood parasites, this bird neither builds nests nor displays parental care, effectively preventing vertical conspecific transmission from biological parents to chicks ($q_{vc} = 0$). This is a generalist bird parasitizing more than 90 different passerine species in 17 families in South America and beyond[41–43]. Given that *M. bonariensis* is a brood parasite, one would expect that this bird might have (i) specific, single-host mite species that co-diverged with their hosts over a long evolutionary time and (ii) foster parent mite species whose coevolutionary history has been mostly driven by host shifts rather than codivergence events. *Molothrus*-specific mites (i.e., single-host mites consistently found only on *M. bonariensis*) can be transferred only by horizontal contact between conspecific hosts, i.e., other *M. bonariensis* (Fig. 1A: $q_{hc}$). In contrast, *Molothrus*-alien mites (i.e., mites found both on *M. bonariensis* and one or more species of its foster parents) would be transferred mostly via vertical interspecific care (Fig. 1A: $q_{vi}$), and, potentially to a much lesser extent, by horizontal interspecific interaction (Fig. 1A: $q_{hi}$; see justification in the section "*Molothrus*-specific mites have higher species richness …"). At the macroevolutionary timescale, the constant transmission of mites from foster parents likely has provided more opportunities for host switches than what would have been expected in non-brood parasite bird-mite systems. The *M. bonariensis* system, therefore, can be used to evaluate the magnitude of horizontal conspecific transmission ($q_{hc}$) vs. interspecific transmissions (i.e., vertical $q_{vi}$, and horizontal $q_{hi}$) and their influence in long-term co-evolution in host-symbiont systems.

Here we accomplish the above goal by quantifying rates of different types of short-term transmission that reflect host specificity and comparing coevolutionary patterns among multiple mite lineages (e.g., *Proctophyllodes*, *Amerodectes*, *Trouessartia*) that offer independent biological replicates in the *Molothrus* system. Specifically, we evaluate the effective horizontal versus vertical mite transmission rates assuming that each mite species on each *M. bonariensis* specimen resulted from at least a single successful host switch event. The distribution of mites in each host specificity category, therefore, should reflect the relative conspecific and interspecific transmission rates. Then we used event-based cophylogenetic reconciliation analyses to estimate the number of four coevolutionary events (codivergence, duplication, host switch, extinction) that occurred in this system on the macroevolutionary scale. We discuss the implications of our results to explain the observed macroevolutionary and microevolutionary patterns in this system.

Our null hypothesis is that if horizontal conspecific transmission ($q_{hc}$) is lower than interspecific transmission (rates $q_{vi}$ and $q_{hi}$) in the *Molothrus* system, then both host specificity and cophylogenetic congruence should be low due to a high frequency of interspecific transmission from foster parents ($q_{vi}$) and the absence of conspecific vertical transmission ($q_{vc}$) (Fig. 1B: H_0). Otherwise, if conspecific transmission is higher than interspecific (H_1: $q_{hc} > q_{vi} + q_{hi}$), host specificity should be high, while the level of cophylogenetic congruence cannot be predicted because many host switch opportunities may arise over a long evolutionary time. Hypothesis H_1 (Fig. 1B: H_1) favors a greater role of horizontal transmission in maintaining host specificity than it is assumed in the literature, e.g.,[25,39]. Furthermore, these two hypotheses (H_1 and H_0, respectively) can also answer the question of whether or not host-specific symbionts can persist via the horizontal transmission route alone.

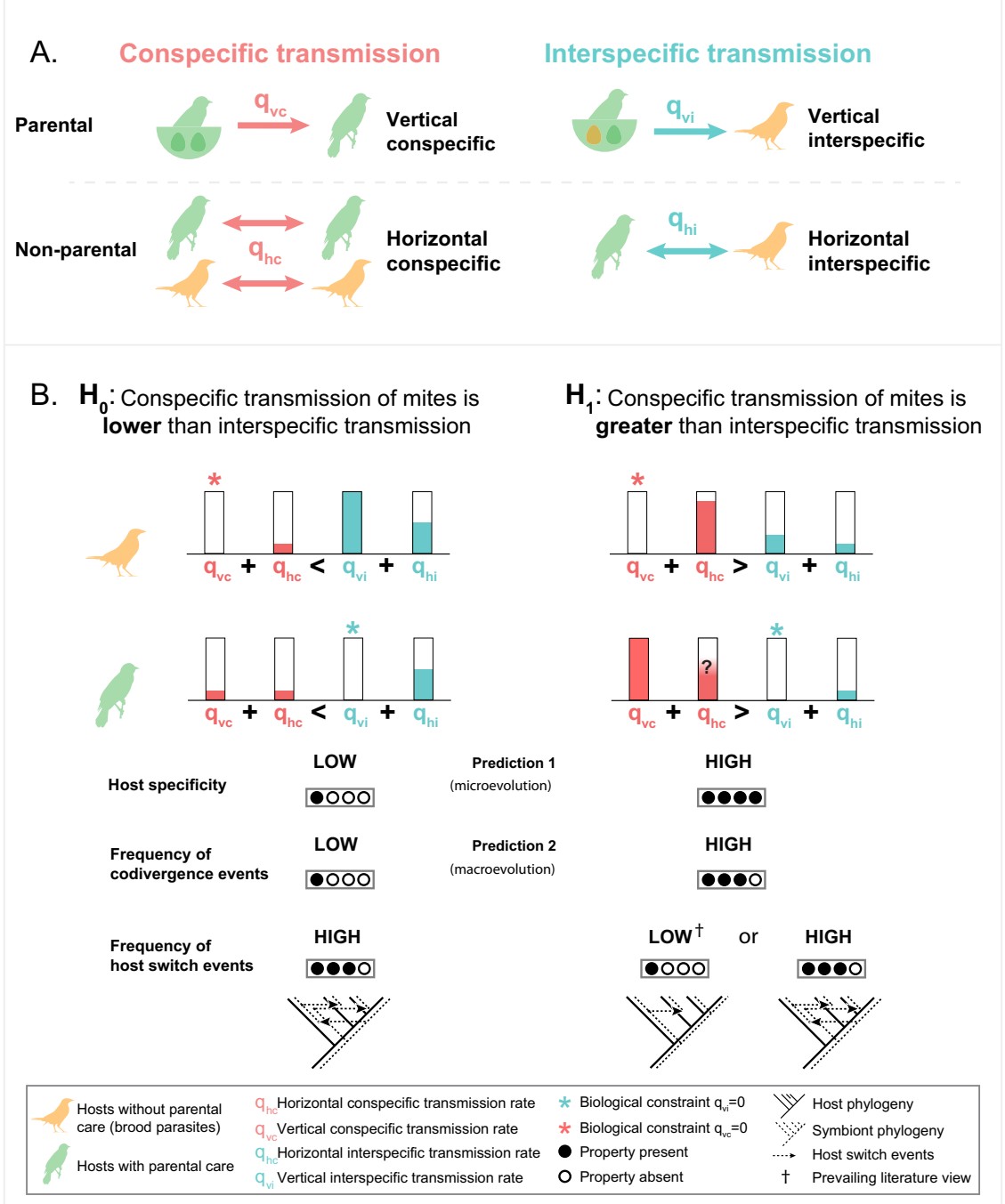

**Fig. 1 Influence of transmission modes on host specificity and cophylogenetic congruence in feather mite–bird systems. A** Conspecific (vertical/horizontal) vs interspecific (vertical/horizontal) transmissions in different bird systems (brood parasites and non-brood parasites); **B** two hypotheses on how different symbiont transmission types may affect the mite host specificity and cophylogenetic congruence with their avian hosts. Our null hypothesis ($H_0$) proposes that a lower conspecific transmission rate (in comparison to the interspecific rate) results in lower host specificity, lower number of codivergence events, and a higher number of host switches; whereas a higher rate of conspecific transmission ($H_1$) is expected to result in higher host specificity, higher number of codivergence events, and lower number of host switches. Artwork by L.G.A.P.

## Results

### *Molothrus*-specific mites have higher species richness, prevalence, and abundance than *Molothrus*-alien mites. To identify *Molothrus* host-specific mites and mites primarily associated with *Molothrus* foster parents, we conducted an extensive mite survey focusing on *M. bonariensis* (144 specimens). To aid in identifying and classifying mites into the *Molothrus*-specific vs alien categories, we also sampled mites from its common foster parents (27 species, 69 specimens) (Supplementary Data 1 and 2) and considered the literature

data. Out of the 144 *M. bonariensis* specimens inspected for feather mites, 5 individuals entirely lacked mites, and 139 had mites. Mites were identified and assigned to three categories, *Molothrus*-specific, *Molothrus*-alien, and quill-and-skin mites QSM; with the former two categories representing feather vane mites lacking a vector transmission, while the latter category representing rare ecological groupings, some of which can be transmitted by a vector (see Methods). *Molothrus*-specific mites (5 species) were exclusively and consistently found on 130 *M. bonariensis* specimens with moderate

**Table 1 Feather mite species identified on *Molothrus bonariensis*.**

| Mite ($n = 29$) | Prevalence by host individual (%) | Proportion of mite records, p/a (%) | Mite records, p/a ($n = 365$) | Mite abundance ($n = 1246$) | Mite abundance (%) | DNA Data |
|---|---|---|---|---|---|---|
| ***Molothrus*-specific** | | | | | | |
| *Amerodectes molothrus* | 60.42 | 23.84 | 87 | 355 | 28.49 | Yes |
| *Mesalgoides* sp. 1 | 30.56 | 12.05 | 44 | 122 | 9.79 | |
| *Proctophyllodes molothrus* | 59.72 | 23.56 | 86 | 344 | 27.61 | Yes |
| *Trouessartia* sp. 6 | 13.19 | 5.21 | 19 | 49 | 3.93 | |
| *Xolalgoides* sp. 1 | 25.69 | 10.14 | 37 | 115 | 9.23 | |
| **Subtotal** | | **74.8** | **273** | **985** | **79.0** | |
| ***Molothrus*-alien** | | | | | | |
| *Amerodectes bilineatus* | 0.69 | 0.27 | 1 | 5 | 0.40 | |
| *Analges* sp. 1 | 1.39 | 0.55 | 2 | 33 | 2.65 | |
| *Analges* sp. 5 | 0.69 | 0.27 | 1 | 7 | 0.56 | |
| *Analges* sp. 6 | 1.39 | 0.55 | 2 | 11 | 0.88 | |
| *Analges ticotico* | 0.69 | 0.27 | 1 | 6 | 0.48 | |
| *Mesalgoides* sp. 2 | 7.64 | 3.01 | 11 | 14 | 1.12 | |
| *Mesalgoides* sp. 3 | 2.78 | 1.10 | 4 | 7 | 0.56 | |
| *Platyacarus* sp. | 1.39 | 0.55 | 2 | 12 | 0.96 | |
| *Proctophyllodes* aff. *atyeoi* | 2.78 | 1.10 | 4 | 16 | 1.28 | |
| *Proctophyllodes* cf. *thraupis* | 2.78 | 1.10 | 4 | 12 | 0.96 | |
| *Proctophyllodes* sp. 16 | 1.39 | 0.55 | 2 | 7 | 0.56 | |
| *Proctophyllodes* sp. 4 | 2.78 | 1.10 | 4 | 8 | 0.64 | |
| *Proctophyllodes* sp. 5 | 4.17 | 1.64 | 6 | 13 | 1.04 | |
| *Proctophyllodes carmenmirandae* | 3.47 | 1.37 | 5 | 9 | 0.72 | |
| *Trouessartia* aff. *megaplax* | 1.39 | 0.55 | 2 | 13 | 1.04 | |
| *Trouessartia capensis* | 6.94 | 2.74 | 10 | 18 | 1.44 | |
| *Trouessartia* sp. 7 | 2.08 | 0.82 | 3 | 10 | 0.80 | |
| *Trouessartia* cf. *sicaliae* | 0.69 | 0.27 | 1 | 2 | 0.16 | Yes |
| *Xolalgoides* sp. 2 | 2.78 | 1.10 | 4 | 11 | 0.88 | |
| **Subtotal** | | **18.9** | **69** | **214** | **17.2** | |
| **QSM** | | | | | | |
| *Dermationidae* | 4.17 | 1.64 | 6 | 10 | 0.80 | |
| *Dermoglyphus* cf. *passerinus* | 5.56 | 2.19 | 8 | 16 | 1.28 | |
| *Microlichus* cf. *americanus* | 2.78 | 1.10 | 4 | 4 | 0.32 | |
| *Metamicrolichus* cf. *phasianus* | 1.39 | 0.55 | 2 | 7 | 0.56 | |
| *Strelkoviacarus brasiliensis* | 2.08 | 0.82 | 3 | 10 | 0.80 | |
| **Subtotal** | | **6.3** | **23** | **47** | **3.8** | |
| **Total** | 144 | 365 | 365 | 1246 | 1246 | 3 |

Prevalence based on the mite presence/absence data (p/a), abundance, and molecular data availability of feather mites from *M. bonariensis*. Species are grouped by host specificity categories.

to high prevalence (13–60%, Table 1: *Proctophyllodes molothrus*, *Amerodectes molothrus*, *Xolalgoides* sp.1, *Mesalgoides* sp.1, and *Trouessartia* sp.6)[44,45]. *Molothrus*-alien mites (19 species, Table 1) were recorded on few host individuals (prevalence 0.69–7.5%). Twenty-six out of the 69 *Molothrus*-alien records could be attributed to known *M. bonariensis* foster parents based on 8 mite species identified with morphological and molecular evidence (Supplementary Note 5, Supplementary Data 1). Lastly, QSM mites (5 species, Table 1) were similar to the *Molothrus*-alien category as these mites were also present at low prevalence (1.4–5.6%) (Table 1). Cumulative mite abundance (the total number of all mite specimens in each category) was also higher in *Molothrus*-specific mites in comparison with the two other categories (Table 1).

The co-occurrence pattern of *Molothrus*-specific and *Molothrus*-alien or QSM mites was also evidenced by our quantitative data: (i) in 52.5% of cases, only *Molothrus*-specific mites were found on a particular host individual; while (ii) *Molothrus*-specific mites were more prevalent (34.5%) than either *Molothrus*-alien (16.4%) or QSM (6%) in co-occurring mite category records (Table 2); and (iii) only in 2.5% of cases, mites originating solely from the foster parents ($q_{vi} + q_{hi}$) were found (Table 2).

These data provide evidence for the low rate of horizontal interspecific transmission (Fig. 1: $q_{hi}$).

**The rate of horizontal social symbiont transmission is high.** Based on the prevalence and co-occurrence pattern of mites, we inferred that the effective conspecific transmission rate is around 3.9 times the interspecific transmission rate (74.8% vs. 18.9%, Table 2). Since *Molothrus*-specific mites can only disperse between *M. bonariensis* via horizontal conspecific transmission ($q_{hc}$) and *Molothrus*-alien mites predominantly disperse via vertical interspecific transmission from foster parents ($q_{vi}$), it can be inferred that cowbird-to-cowbird transmission is at least 3.9 times more frequent than foster parent transmission in this system. Because the QSM group can disperse either from foster parents to *M. bonariensis* or horizontally (by host social contacts or via phoresy), we cannot determine if they belong to conspecific or interspecific transmissions. Assuming that QSM mites disperse either only via (foster) parental care or only via host horizontal contacts, we estimate the overall ratio between *Molothrus*-to-*Molothrus* and foster parent transmission in the *M. bonariensis*

**Table 2 Transmission in three host specificity categories of mites associated with *Molothrus bonariensis*.**

| Type of mites | Host individuals | *Molothrus*-specific (5 species) | *Molothrus*-alien (19 species) | QSM (5 species) |
|---|---|---|---|---|
| Single type of mites | 81 (56.3%) | | | |
| *Molothrus*-specific mites only | 73 (50.7%) | 147 (40.3%) | n/a | n/a |
| *Molothrus*-alien mites only | 7 (4.9%) | n/a | 9 (2.5%) | n/a |
| QSM only | 1 (0.7%) | n/a | 0 | 1 (0.3%) |
| Co-occurrence of different types of mites | 58 (40.3%) | 126 (34.5%) | 60 (16.4%) | 22 (6%) |
| *Molothrus*-specific+*Molothrus*-alien+QSM | 7 (4.9%) | 17 (4.6%) | 11 (3%) | 7 (1.9%) |
| *Molothrus*-specific+*Molothrus*-alien | 36 (25%) | 73 (20%) | 48 (13.1%) | n/a |
| *Molothrus*-specific+QSM | 14 (9.7%) | 36 (9.9%) | n/a | 14 (3.8%) |
| *Molothrus*-alien+QSM | 1 (0.7%) | n/a | 1 (0.3%) | 1 (0.3%) |
| **Horizontal conspecific transmission ($q_{hc}$)** | | **273 (74.8%)** | n/a | n/a |
| **Vertical/Horizontal interspecific transmission ($q_{vi}+q_{hi}$)** | | n/a | **69 (18.9%)** | n/a |
| **Undetermined type of transmission** | | n/a | n/a | **23 (6.3%)** |
| Horizontal conspecific ($q_{hc}$) + Vertical interspecific ($q_{vi}$) + Horizontal interspecific ($q_{hi}$) (transmission events total) | 139 (96.5%) | 273 (74.8%) | 69 (18.9%) | 23 (6.3%) |

Transmission events estimated from symbiont occurrence data for the three mite specificity categories (n/a = not applicable). For host individuals, counts and percentages are given. For the mite host specificity categories, values are counts (percentages) of unique host–symbiont records (=percentage of cases for each category). Data are summarized from 365 mite records (=total of transmission cases) sampled from 144 bird individuals. Of them, 139 bird individuals had mites, 5 bird individuals lacked any mites. Host counts may overlap since a host individual may harbor mites from different host-specificity categories.
Different types of transmission modes and their estimated rates are highlighted in bold.

system is within the interval of 3.0–4.3 (74.8/25.2–81.1/18.9) (Table 2), where the lower value (3.0) is a conservative estimate of horizontal conspecific vs. vertical interspecific transmission rate. Thus one can conclude that *Molothrus*-to-*Molothrus* transmission ($q_{hc}$) is at least three times greater than foster parent transmission ($q_{vi}$) and horizontal interspecific transmission ($q_{hi}$), i.e., $q_{hc} > (q_{vi} + q_{hi}) = 3:1$.

**Molothrus-associated mites are sister taxa to mites associated with other *Molothrus* species or foster parents phylogenetically.** In order to understand macroevolutionary patterns between mites and *Molothrus*, we obtained molecular data from 118 specimens of field-collected mites associated with *Molothrus* and/or putative foster parents (29 bird species from 21 genera and 10 families, Genbank accessions CO1: MW814590–MW814707 and HSP70: MW829221–MW829276; see Table 1, Supplementary Data 2, Supplementary Note 3). We reconstructed mite phylogenies using 6 loci by supplementing our dataset with 153 mite sequences and 127 hosts from previous publications[5,6] (Fig. 2). The sampling of mite phylogeny encompasses ca. 48.6% of the total named mite species for the families Proctophyllodidae and Trouessartiidae ($n = 253$), and ca. 18.8% of the total host species ($n = 425$) (considering New World Passeriformes records only and unique mite species haplotypes)[46]. Out of the 29 *M. bonariensis*-associated mite species, we provided phylogenetic information for two *Molothrus*-specific and one *Molothrus*-alien species (highlighted by orange in Fig. 2) and identified their sister taxa. They belong to three families/subfamilies of mites—Pterodectinae, Proctophyllodinae, and Trouessartiidae, and therefore represent three independent host-symbiont coevolutionary histories (Fig. 2).

The sister taxa of two of the three mite species, *Amerodectes molothrus* and *Proctophyllodes molothrus*, are mites from *M. ater* (*M. bonariensis*' sister species) from Mexico (A, B in Figs. 2 and 3). A species delimitation analysis based on genetic distances within and between the sister pairs, *Proctophyllodes* sp. ex. *M. ater - Proctophyllodes molothrus* (CO1 K2P = 5.5%, intraspecific distance for *P. molothrus* 0.2–0.3%) and *Amerodectes tretiakae - Amerodectes molothrus* (CO1 K2P = 9.5%, intraspecific distance for *A. molothrus* = 0.5%) (Fig. 3A, B), unambiguously placed these four genetically distinct OTUs as separate species (Assemble Species by Automatic Partitioning (ASAP) algorithm score = 5.50). The third species, *Trouessartia sicaliae*, shows similarity to mites

from putative foster parents, *Sicalis flaveola* and *S. luteola* (Thraupidae) (CO1 K2P = 3.6–6.5%) (Fig. 3C). Another *M. ater*-associated mite species, *Proctophyllodes egglestoni*, displays a similar pattern in that it is closely related to mites from *Agelaius phoeniceus* (CO1 K2P = 0.2%), a common foster parent of *M. ater*[43] (Fig. 3D).

**Host switches and codivergences occur at nearly the same frequencies.** For the three mite lineages, A, B+C, and D (Fig. 3), we performed separate cophylogenetic analyses using dated host and parasite phylogenies (Fig. 3, Supplementary Note 4, Supplementary Figs. 1–5) to identify either ongoing (with gene flow) or historical host switching (no gene flow). For all these lineages, a significant phylogenetic congruence was detected, $p < 0.01$ (see Methods, Supplementary Note 4, Supplementary Figs. 1–5). Our time estimates provide a temporal context for mite-bird colonization events (Fig. 3) and suggest that host switches were temporally possible with respect to host diversification. For the three *Molothrus*-associated mite lineages, we recovered 2 codivergences and 3 host switch events. Two codivergence events represent speciation that occurred within the *Molothrus* genus, between *Amerodectes molothrus*—*A. tretiakae* (Fig. 3A) and *Proctophyllodes molothrus*—*P.* sp. (Fig. 3B). However, host switches occurred at their ancestral nodes: in *Amerodectes*, there was a host switch from ancestral *Molothrus* to ancestral *Sicalis* (Fig. 3A); in *Proctophyllodes*, there was 1 host switch from an unknown host to ancestral *Molothrus* (Fig. 3B); and in *Trouessartia*, *T. sicaliae* switched its host from *Sicalis flaveola* to *M. bonariensis* (Fig. 3D). *Proctophyllodes egglestoni*, on the other hand, has experienced ongoing transmission from foster parents to *M. ater*, possibly through both vertical and horizontal routes as some foster parent species may also form mixed flocks with *M. ater*[47] (Fig. 3C). Below we present a detailed description of topology- and time-informed coevolutionary scenarios in the cowbird system (Fig. 3A–D).

*Amerodectes molothrus complex.* A putative host switch onto ancestral *Molothrus*, followed by host switch of *Amerodectes* mites from ancestral *Molothrus* to their foster parents (ancestral *Sicalis*) (Fig. 3A: $\tau_2$), which resulted in their codivergence on *Molothrus* and possibly on *Sicalis* (Fig. 3A: $\tau_2$–$\tau_0$). This scenario is evidenced in all our cophylogenetic reconciliations and by the fact that mites from *Sicalis* and *Molothrus* are monophyletic (Fig. 2A).

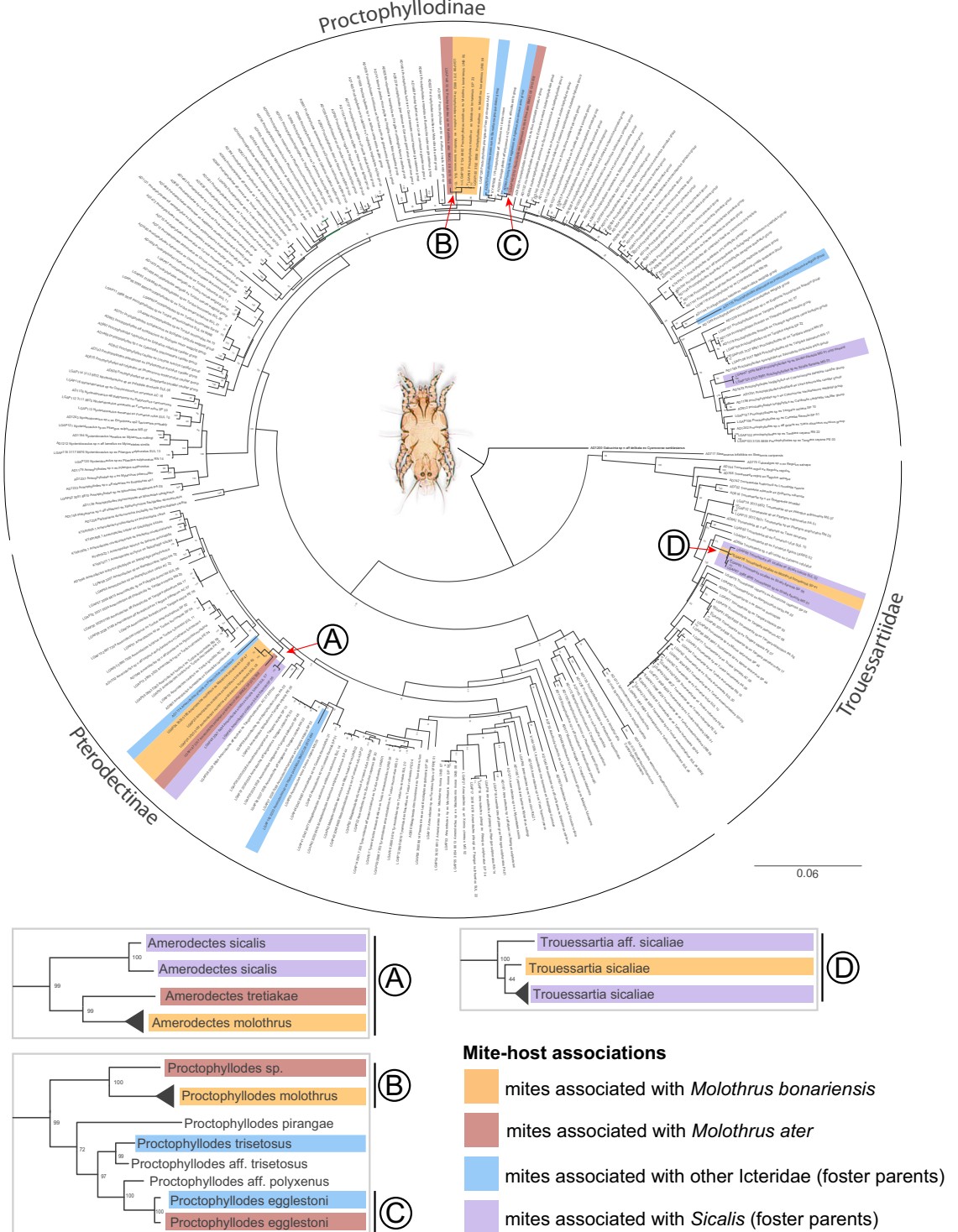

**Fig. 2 Phylogenetic relationships of feather mites showing four independently evolved lineages associated with _Molothrus_.** Maximum Likelihood phylogeny of feather mites based on six genes (5 nuclear, 1 mitochondrial). Nodal support higher than 75% (estimated by ultrafast bootstrap with 1000 replicates) is shown by thicker lines. The major mite lineages, Trouessartiidae, Proctophyllodinae and Pterodectinae, are highlighted. Mites from cowbirds (_Molothrus bonariensis_ and _M. ater_) and their foster parent birds: finches (_Sicalis flaveola_ and _S. luteola_) and other Icteridae are highlighted. Portions of this phylogeny, exemplifying important host shifts to _Molothrus_ are given for each mite lineage on insets (**A**–**D**); these cases are further considered in detail in Fig. 3. Artwork by L.G.A.P.

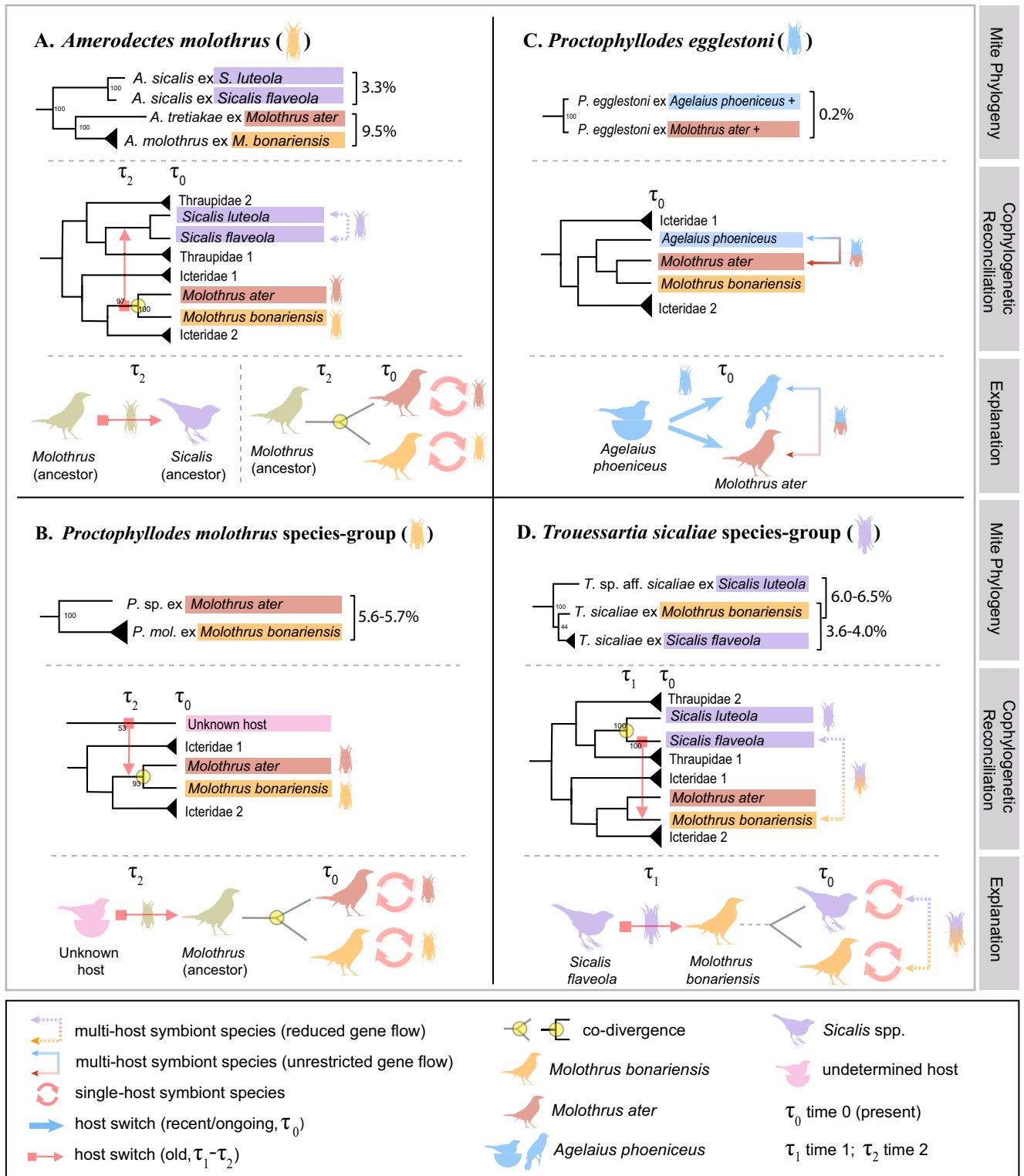

**Fig. 3 Cophylogenetic scenarios of feather mites associated with *Molothrus*. A–D** Mite molecular phylogenetic trees, cophylogenetic reconciliation analysis, and schematic explanations of cophylogenetic scenarios (for complete phylogenies, see Fig. 2; and Supplementary Figs. 1–5). Nodal support is given for each branch (ultrafast bootstrap, 1000 replicates). Molecular CO1 K2P genetic distances are given as percentages. See the Result section for a detailed explanation of cophylogenetic scenarios. Artwork by L.G.A.P.

Our host-based calibration time estimates for this host switch make it temporally possible (7.9 Mya), contrarily to our fossil mite calibration (15.5 Mya), given that it occurred before the origin of both *Molothrus* and *Sicalis* (Supplementary Figs. 1 and 2). Our divergence time estimates of mites from *Molothrus ater*

and *M. bonariensis* (4.4/3.7 Mya, host/fossil calibration) predate the divergence time estimates of their hosts (i.e., ~1 Mya), but are close to the extreme value (3.8 Mya) from the literature[48–50]. In contrast, *Amerodectes* mites associated with *Sicalis flaveola/luteola* originated after their hosts split into species (0.6/

1.2 Mya vs 4.8–5.6 Mya) (Supplementary Fig. 1, 2). Taken together, our cophylogenetic evidence and time estimates suggest that in the *Amerodectes molothrus* complex, a host switch occurred from the ancestral *Molothrus* to ancestral *Sicalis*.

**Proctophyllodes molothrus complex.** A putative host switch onto ancestral *Molothrus*, followed by codivergence on *Molothrus* (Fig. 3B; $\tau_2 - \tau_0$). Because this group is sister to the mite lineage having hosts from many families (Cardinalidae, Icteriidae, Passerellidae, and Icteridae) (Fig. 2B), its ancestral host cannot be identified with certainty. Our time estimates for this host switch (1.9–5.0/3.7–9.6 Mya host/fossil calibration, respectively) are plausible, given that they are completely (former) or partially (latter) within the timing of origin of *Molothrus*.

**Proctophyllodes egglestoni.** Ongoing mite transmission from foster parents and possibly by social contact (Fig. 3C; $\tau_0$). Mites from *Molothrus ater* clustered with those of the red-winged blackbird, *Agelaius phoeniceus*, and their CO1 genetic distances were nearly identical (0.2%) (Figs. 2C and 3C). The high sequence identity is indicative of an ongoing mite transmission from foster parent to *Molothrus* and/or via social contacts in mixed flocks. This is a multihost mite, mostly associated with icterids[47].

**Trouessartia sicaliae complex.** Likely host switch to *M. bonariensis* from foster parents followed by on-host divergence (Fig. 3D; $\tau_1 - \tau_0$). One DNA sequence of *Trouessartia* from *M. bonariensis* was recovered as nested within the *Trouessartia sicaliae* morphospecies associated with thraupids *Sicalis* spp. (Figs. 2D and 3D), which are known foster parents of *M. bonariensis*. However, the *Trouessartia* lineage from *M. bonariensis* had substantial CO1 genetic distance (3.6%), indicating that this *Sicalis* to *Molothrus* switch occurred sometime in the past, or is a result of a transfer from an unsampled host. Ongoing gene flow via conspecific social horizontal transmission occurring at low rate cannot be excluded.

## Discussion

To investigate the magnitude of host-specific symbiont transmission via horizontal conspecific contact ($q_{hc}$) vs interspecific transmissions ($q_{vi}$ and $q_{hi}$), we studied a generalist brood parasitic passerine, the shiny cowbird *M. bonariensis*, and its obligatory symbiotic feather mites. Despite the lack of parental care in the host and the absence of regular parent-to-offspring mite vertical transmission ($q_{vc} = 0$), five mite species (*Molothrus*-specific) were able to colonize new host generations horizontally via host social conspecific contact ($q_{hc}$). In contrast, mites originating from the foster parents (*Molothrus*-alien) and/or dispersing through other modes (QSM) had a higher species richness (24 species, 11 genera), despite their overall low prevalence. This is an expected outcome because *M. bonariensis* has a broad range of foster parent hosts ($n = 97$)[43], which contributes to the great diversity of mites in the *Molothrus*-alien category.

Based on the known biology of *M. bonariensis*, social transmission of *Molothrus*-specific mites is only possible via horizontal conspecific contact between bird individuals (Fig. 1: $q_{hc}$), which can occur when their fledglings leave their foster parent's nest for the formation of foraging and roosting flocks, during courtship, copulation, and other conspecific interactions[51–56]. *Molothrus*-specific mites do not occur on any other bird species as shown by our extensive survey and previous research[44,45,57]. Thus, there is no evidence that these mites are transmitted via foster parental care or that *M. bonariensis* foster parents can serve as vectors of *Molothrus*-specific mites. Furthermore, horizontal interspecific transmission ($q_{hi}$) should be minimal in all cases as *alien*-only mites were found in only 2.5% of *M. bonariensis* individuals vs specific-

only in 40.3% (Table 2); and even this small figure (2.5%) can be fully or partially attributed to foster parent transmission. Our estimation $q_{hc} > (q_{vi} + q_{hi}) = 3:1$ (see above) supports our alternative hypothesis (Fig. 1: $H_1$), suggesting that conspecific horizontal transmission of host-specific mites ($q_{hc}$) is higher than vertical interspecific transmission and horizontal interspecific transmission ($q_{vi} + q_{hi}$ and $q_{vi} > > q_{hi}$). It is also possible that mites transmitted from foster parents are replaced by *Molothrus*-specific mites through competitive exclusion. Unfortunately, here we cannot provide direct evidence for this hypothesis because samples of immature bird stages were not available for study. Under an alternative scenario, when horizontal interspecific transmission of *Molothrus*-alien mites is equal to or greater than horizontal conspecific transmission of *Molothrus*-specific mites (i.e., $q_{hi} \geq q_{hc}$), hosts harboring *alien*-only mites would be expected to be found at the same or greater frequency as hosts harboring only *Molothrus*-specific mites. However, this is not the case in this system.

In the cowbird system, all *Molothrus*-specific mites were transmitted horizontally via conspecific contacts, which likely occurred gradually with host age as the birds experienced an increasing rate of social contact. This has been shown in feather mites associated with the Australian bushturkeys (Galliformes: Megapodiidae), a host that has minimal parent-to-offspring contact (eggs are buried, young birds are fully fledged on hatching)[58]. In *M. bonariensis*, horizontal conspecific transmission has maintained host specificity of its mite symbionts and also the continuity of the mites' generations, resulting in 90.3% prevalence of *Molothrus*-specific mites among hosts carrying any mites. On a macroevolutionary scale, the coevolutionary patterns of *Molothrus*-associated mites, however, exhibit a range of possible events, from strict codivergence to incomplete and complete host switches, providing snapshots of how a low rate of interspecific transmission on a macroevolutionary timescale could result in potential host switch events. Our coevolutionary analyses revealed a pronounced pattern of host switching and codiversification. As an example, the two most common and abundant mite species, *Amerodectes molothrus* and *Proctophyllodes molothrus*, were involved in historical host switches (Fig. 3A, B). Subsequently, these two newly established mite lineages have become genetically isolated and specific to their new hosts. For multihost mites, failure to speciate with ongoing gene flow might be more common. For example, an exceptional multihost generalist mite, *Proctophyllodes egglestoni*, can probably be transferred from *M. ater* to its foster parents (*Agelaius phoeniceus*) and back through aggressive (e.g., nest defense) and gregarious (e.g., mixed-flocks) behaviors[59]. These ongoing transmissions are supported by the shallow genetic distances between the mites from the two hosts, COX1 K2P = 0.2% (Fig. 3D). However, mites using this strategy were not found in the *Molothrus bonariensis* system, thus allowing us to accurately quantify the rate of foster parent transmission in *M. bonariensis*.

Previous studies have suggested the presence of horizontal conspecific and vertical interspecific transmission in cowbirds and other brood parasitic birds by the presence of host-specific or parent-specific symbionts, albeit with no quantitative data or phylogenetic analyses[57,60–63]. Furthermore, even in non-brood parasitic systems, horizontal conspecific transmission has also been observed as the main transmission route of symbionts, such as feather mites from red-billed choughs[64], barn swallows[65], and in feather lice from bee-eaters[64–66]. Thus, our data on the high rate of social (horizontal) transmission in the *Molothrus* system potentially represent a more general pattern in feather mites that needs to be further investigated in other bird systems as well.

In host-symbiont systems, the general expectation is that parent-offspring vertical transmission results in both cophylogenetic congruence and high host specificity[7,14,16,23,67]. However, in the

feather mite-bird symbiotic system, low cophylogenetic congruence and relatively high host specificity have been simultaneously detected in the same systems[3,5,6,32]. If we assume that most feather mites disperse vertically[25,39,68], only the high host specificity can be explained; the low cophylogenetic congruence cannot be explained in terms of a larger contribution of predominant vertical transmission. Our work may reconcile these opposing observations by (i) estimating relative rates of ongoing horizontal and foster parent vertical transmission (microevolutionary scale) and (ii) quantifying codivergence vs non-codivergence coevolutionary events (macroevolutionary scale). The high host specificity observed in the *Molothrus*-specific mites can be therefore explained solely by the high rates of horizontal conspecific transmission, despite the lack of vertical conspecific transmission. On the macroevolutionary scale, although brood parasitic birds have $q_{hc}$ larger than $q_{vi}$, the frequency of host switches cannot be extrapolated from these rates directly (Fig. 1B: H$_1$). Here we observe an equal number of codivergence vs host-switch events. We therefore suggest that on a microevolutionary scale, both vertical ($q_{vc}$, $q_{vi}$) and horizontal transmissions ($q_{hc}$, $q_{hi}$) can affect host specificity of feather mites, while these transmissions may not substantially influence their coevolutionary scenarios on the macroevolutionary scale. For example, the probability of a successful host switch, an event affecting host-symbiont coevolution, may depend on multiple factors other than transmission rates, such as the competitive abilities of a particular symbiont species, common resources shared by unrelated hosts, and niche availability[16,69,70].

In summary, our work highlights that symbiont horizontal transmission via conspecific social contacts is an important dispersal mode of *Molothrus*-specific mites onto new host individuals in the *Molothrus*-feather mite system. This horizontal transmission alone (without parent-offspring vertical transmission) can maintain highly abundant, dominant, and host-specific species of obligate mite symbionts on their hosts. Here, we identified five independently evolved lineages of mites specific to *M. bonariensis* that colonize new generations of hosts exclusively through horizontal route of transmission. These symbionts persist on *M. bonariensis* at high prevalence and abundance despite the constant influx of new diverse mite colonists transmitted vertically from over 90 species of foster parents. On average, mite species dispersing via conspecific horizontal contacts are three times more likely to colonize *M. bonariensis* than mites transmitted vertically via foster parents. Our data therefore provide evidence challenging the traditional view of the importance of vertical transmission as the main force generating host specificity on a microevolutionary scale[23]. On a macroevolutionary scale, we show that horizontal transmission maintained these *Molothrus*-specific mites on their hosts over a long evolutionary time, at least 1.38 Mya since the split of *M. bonariensis* and *M. ater*. There were both codivergence and host switch events, occurring nearly at the same frequencies. This suggests that macroevolutionary patterns, which are based on rare coevolutionary events, cannot be easily generalized from short-term evolutionary trends, such as transmission mode and rates.

## Methods

**Taxon sampling**. We sampled 144 *M. bonariensis* specimens for feather mites in Brazil, 22 captured in the wild (by plucking infested feathers), 10 roadkill donated bodies (by washing), and 112 museum dry skins (by feather-ruffling) (see Supplementary Notes 1 and 2 for sampling details, and Supplementary Data 1 for shiny cowbird mite list). We have complied with all relevant ethical regulations for animal use (permits MMA 57944-3, CEUA 12/2017). For museum samples, to exclude potential cross-contamination, confidence scores were applied as detailed in

Supplementary Note 2. This dataset was used to estimate the mite transmission rates. To identify foster parent mite species and classify mites into *Molothrus*-specific and *Molothrus*-alien categories, we also sampled mites from (i) putative *Molothrus bonariensis* foster parent species:[43] 68 specimens (27 species, 19 genera, 10 families) captured with mist nets from different regions in Brazil, and (ii) 1 specimen of *Molothrus ater*, which is sister to *M. bonariensis* (see supplementary file P21.MCC.tre in ref. [71]), from Mexico (Supplementary Data 2: column "Data description"). At least one representative of each mite morphospecies of the two common subfamilies Proctophyllodinae, and Pterodectinae, and the family Trouessartiidae, was chosen per bird species and per region for DNA sequencing (Supplementary Data 2). A total of 118 mite specimens were sequenced for two genes (174 new sequences); in addition, 153 previously deposited GenBank sequences of 153 mite species from 127 bird species (90 genera, 36 families) were included in the analyses as well (Supplementary Data 2).

**Definition of host specificity categories**. Using morphological data (from museum skins, dead birds, and field-captured mites, 365 mite records: 29 mite morphospecies, 12 genera, 8 families from 144 *M. bonariensis*; Table 1; Supplementary Data 1; Supplementary Note 1 and 2), we identified mite species and assigned them in three categories (*Molothrus*-specific, *Molothrus*-alien and Quill-and-skin QSM mites) based on the following criteria: exclusivity of an association with *M. bonariensis* (i.e., a single-host mite species occurring only on *M. bonariensis*)[72], phylogenetic relationships and genetic distances as compared to mites from foster parents or one of its sister brood parasitic species, *Molothrus ater*[71], and transmission types:

(i) *Molothrus*-specific—mite species found exclusively on *Molothrus bonariensis*; in molecular trees they should be distantly related to mites known from foster parent passerines and be closely related to mites from one of its sister host species, *Molothrus ater*.

(ii) *Molothrus*-alien (foster parent mites)—found principally on *M. bonariensis* foster parent birds and also they can be found on *M. bonariensis* (likely as a result of transmission from foster parents); in phylogenetic trees, they should be closely related (i.e., have zero length or shallow branches) to mites specific to a particular foster parent passerine species.

(iii) Quill-and-skin mites (QSM)—inhabit the host skin (e.g., *Microlichus*, *Metamicrolichus*, *Strelkoviacarus*) or live inside feather quills (*Dermoglyphus*). These mites are grouped together because of their restricted contact transmission in comparison to that of plumage feather mites (groups i and ii). For example, when two bird individuals come into contact, the likelihood of sharing skin or quill mites is likely to be much lower than that of sharing plumage mites. Furthermore, skin mites are naturally rare and have low prevalence and abundance[73] in comparison with plumage mites[74]. Some skin-inhabiting mites also have the ability to be horizontally transferred across interspecific and intraspecific hosts via phoresy on louse flies (Diptera: Hippoboscidae)[36,75,76]. As a result, in terms of their transmission patterns, these mites cannot be directly compared to plumage-inhabiting feather mites (groups i and ii).

**DNA Amplification and Sequencing**. We sequenced the mitochondrial cytochrome c oxidase subunit I (CO1) gene (1026 nt), a marker that is useful to identify species and phylogenies below the genus level[6,21,77]. To reconstruct deeper mite divergences we considered 5 candidate loci (EF1-α, SRP54, HSP70, 28 S, and 18 S) used in a recent phylogenetic study[5], and selected the

nuclear heat shock protein cognate 5 Hsc70-5 (HSP70) gene (1674 nt) based on its highest Internode Certainty index (IC) value in RaxML 8.2.10[78,79], and amplified this gene for all mites having a unique CO1 haplotype.

We added our sequence data to a large dataset (144 terminals and six genes: 18 S, 28 S, EF1-α, SRP54, HSP70, and CO1; 8546 bp total) generated previously;[5] 9 additional proctophyllodid terminals (CO1-only)[6,80] were also included (Supplementary Data 2). *Gabucinia* (Pterolichoidea: Gabuciniidae) was used as a distant outgroup[5,6]. Our final aligned matrix had 271 terminals, of which 118 were newly sequenced (Supplementary Data 2: GenBank accession numbers, CO1: MW814590–MW814707; HSP70: MW829221–MW829276). See Supplementary Note 3 for details on DNA isolation, amplification, and sequencing.

**Phylogenetic inference**. We inferred a Maximum Likelihood (ML) phylogeny in IQ-Tree 1.6.10[81] under a codon model for the protein-coding genes. Branch support values were estimated by Ultrafast bootstrap with 1000 replicates for the consensus tree[82] (Fig. 2). The best model for each gene partition was estimated in IQ-Tree prior to analyses using the ModelFinder algorithm and corrected Akaike Information Criterion (AICc):[83] 18 S + 28 S (GTR + F + I + G4), EF1-α (KOSI07 + F3X4 + G4), SRP54 (MGK + F1X4 + G4), HSP70 (MG + F3X4 + G4) and CO1 (MG + F3X4 + G4). All these analyses were run using a single command: *iqtree -s data.phy -st CODON5 -alrt 1000 -bb 1000 -nt 7 -p model*. The consensus tree was visualized and edited in FigTree 1.4.4[84]. The tree was rooted using the outgroup genus *Gabucinia* (Pterolichoidea) after the analysis.

**Species delimitation**. For molecular species delimitation, we used the Assemble Species by Automatic Partitioning (ASAP) algorithm[85] and the CO1 locus;[77,86] see[87] for limitations of CO1-only species delimitations. We also evaluate whether putative species form monophyletic lineages by inferring CO1 phylogenies in RaxML 8.2.10 using the GTR + G + I model and 100 bootstrap replicates[79]. We calculated K2P distances in the R package 'ape' v.5.3[88].

**Divergence time estimation**. We performed a Bayesian divergence time estimation in BEAST v2.6.1[89] using a Relaxed Clock Log Normal and the Birth and Death prior model, expecting multiple extinctions in feather mites[5]. Partition schemes and substitution models were found in PartitionFinder v 2.1.1 (GTR + I + G for all partitions, except for the CO1 position 3: GTR + G). For additional details, see Supplementary Note 4. We compared our divergence time estimates with the following known time divergence estimates for the host birds: *Molothrus* (originated 7.4, diversified 4.3 Mya); *Molothrus ater/bonariensis* split 1.0-2.2 Mya; *Sicalis* (originated 9.1 Mya, diversified 7.2 Mya), *Sicalis flaveola/luteola* split 4.8-5.6 Mya[71]. Other published estimates of the *Molothrus ater/bonariensis* split range from 0.8–1.2 to 2.8-3.8 Mya[48–50,90].

**Cophylogenetic analysis**. To quantify the number of different coevolutionary events in the *Molothrus* system, we performed separate parsimony-based reconciliation cophylogenetic analyses in eMPRess[91] for each phylogenetically independent mite lineage (Fig. 2): Pterodectinae (*Amerodectes* A), Proctophyllodinae (*Proctophyllodes* B + C), and Trouessartiidae (*Trouessartia* D). Default event cost values were used as the software automatically selects the average best fitting reconciliations. Symbiont trees were inferred in IQ-Tree 1.6.10 as above (see the 'Phylogenetic Inference' section). Bird phylogenies were obtained from BirdTree[71]. TreeAnnotator v2.6.1 was used to summarize the 1000 host trees into a maximum credibility tree with node heights calculated as median heights. We found phylogenetic information

for all bird species except *Polioptila dumicola*. A uniform OTU naming scheme was used for both molecular and morphological species. Ongoing gene flow was considered to be present between two mite populations from different hosts if their CO1 K2P distances were equal or lower than 5%[77,87].

**Reporting summary**. Further information on research design is available in the Nature Portfolio Reporting Summary linked to this article.

## Data availability

DNA sequences were deposited into GenBank (accession MW814590–MW814707 and MW829221–MW829276). Supplementary Information provide the mite data and host information (Supplementary Data 1); GenBank accession numbers, host and mite collection (Supplementary Data 2); mite collection data and likely foster parents for the two host specificity categories, *Molothrus*-alien and QSM (Supplementary Data 3); Divergence time estimates using well-known host codivergence events (Supplementary Fig. 1) or mite outgroup fossil information (Supplementary Fig. 2); Maximum parsimony cophylogenetic reconciliations for the mite (sub)families: Trouessartiidae, Proctophyllodinae, Pterodectinae (Supplementary Figs. 3–5).

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

## Acknowledgements

We thank the following museum curators for providing access to host skins for our sampling (see Supplemenary Note 2 for museum abbreviations): Dr. Glayson Ariel Bencke (MCN); Dr. Carla Suertegaray Fontana from (MCT); Antenor Silva Junior and Ma. Patrícia Weckerlin e Silva (MHNCI); Dr. Luis Fábio Silveira (MZUSP); Dr. Alexandre Aleixo (MPEG). We thank the following people who helped us organize field trips—Dr. Gerturd Müller (UFPel), Dr. Fabiana F. Bernardon (UFPel), Dr. Lilian Manica (UFPR), Dr. Luiz dos Anjos (UEL), Dr. Edson Guilherme da Silva (UFAC), Dr. Mauro Pichorim (UFRN), Dr. Luciano Naka (UFPE), Dr. Roberto Cavalcanti (UNB), Dr. Sérgio Posso (UFMS), and Dr. Marco A. Pizo (UNESP-RC). Dr. Augusto F. Batisteli donated salvaged shiny cowbirds. Molecular work and analyses were done by L.G.A.P. and P.B.K. at the University of Michigan Museum of Zoology. L.G.A.P. was funded by São Paulo Research Foundation (FAPESP)—2018/21504-0 and 2016/11671-1. P.B.K. was supported by a grant from the Ministry of Science and Higher Education of the Russian Federation within the framework of the Federal Scientific and Technical Program for the Development of Genetic Technologies for 2019-2027 (agreement No. 075-15-2021-1345, unique identifier RF 193021×0012); S.V.M. was supported by the Russian Foundation for Basic Research (RFBR), grant No. 20-04-00500; F.A.H. was supported by National Council for Scientific and Technological Development, CNPq-Brazil Researcher (304479/2019-5).

## Author contributions

L.G.A.P. and P.B.K. performed molecular laboratory work and contributed to writing the manuscript. L.G.A.P. was responsible for the figures' art schemes. F.A.H., Q.H., K.P.J., H.R.B., S.V.M., A.R.P. and B.M.OC. contributed with discussion and critical revision.

## Competing interests

The authors declare no competing interests.
