## [Peer Review File · Communications Biology]

Reviewers' comments:

Reviewer #1 (Remarks to the Author):

Capitalizing on brood-parasite system, in this study Pedroso et al., investigated the importance of vertical and horizontal transmission on evolutionary dynamics between bird hosts and their feather mite symbionts. I find this study very exciting and well written. The study also includes ample amount of data, and analyses. Also, the figure 1 is very clear, which helps to follow the hypotheses and predictions of this paper.

I only have few minor suggestions/comments for this paper:

Line 256: This is very interesting. Have you ever tracked diversity/abundance change of mite species during the ontogeny of cowbirds? It would be cool to know if mites from foster parents are dominant in early in life but later in life they get outcompeted by *Molothrus*-specific mites.

Line 323: It is very nice to use the museum skin collections for such a studies. However, did you find comparable mite abundances in the skin collection and wild-captured or roadkill samples. I am somewhat skeptical when it comes to museum skin collections, which are treated heavily for preservation and might not represent realized mite abundances. If there were any differences, please make a disclaimer about it somewhere.

Line 380: Cophylogenetic analyses: By reading the methods alone I was unable to see if you conducted the cophylogenetic analyses separately for different mite clades (A-D) in the figure 3. Can you provide bit more information on this? Did you run one cophylogenetic analyses or multiple sub cophylogenetic analyses? Also did you do a cophylogenetic analyses on full trees?

Figure 2. Is this figure necessary? For me it is repeating the information in the table 1. But if you decide to keep this figure, can you give colors to each dot based on their species assignment. Then it is easy to see what which two *Molothrus*-specific mite species have the highest abundances.

Figure 3 and 4. I am a colorblind person and it was a bit difficult for me to see the differences in reds and greens. Do you mind changing one of these colours to may be pink? So then everyone can see things better.

Figure 4. This is a complex figure. I think the figure legend need bit more explanation (a summary) on each of the panels, so it would be easier for readers to follow and understand the figure.

Reviewer #2 (Remarks to the Author):

This paper evaluates the magnitude of different mechanisms of parasite transfer from one host species to discuss the implications on the patterns of cophylogeny at macroevolutionary scale and to support the claim in the title: Horizontal transmission maintains host specificity of symbionts in a brood parasitic host.

I color-coded my comments in the pdf attached below with green-highlighted (very good points) and yellow-highlighted feedback (critical points that need to be addressed by the authors) to show that although I found very good points raised and interesting insights, I strongly suggest improving all the criticisms raised. Because the topic is interesting for the community of the evolutionary biologists, I encourage the authors to improve the paper and resubmit it after major revisions.

In general, I suggest improving the presentation of the hypothesis, there are several unclear points, for example the reasoning in lines 82-89 should drive the reader to understand why the H0 was settled as such in line 103, however it is not entirely clear from the perspective of the parasite how the mechanisms of the transfer are linked to the identity of the parasite and the opportunity to switch the host. Overall, I think that the authors must clarify the meaning of "mites forming exclusive associations" and "non-host specific mites" how do we arrive at this categorization and how it is linked to the mechanisms investigated.

I suggest improving the description of the experimental design which seems flawed by the fallacy of unbalanced taxon sampling (1 species/146 specimens versus 27 species/69 specimens). I also suggest the authors to re-organize the Results subsections. There are 5 subheadings, the first one does not really belong to Results section and can be moved in Methods. The following three sub-headings report different metrics to measure the association between mite species and predefined groups. There are several assumptions described in Results section that need to be moved with the presentation of the hypothesis. Moreover, I should warn the authors that most of these results is based on associations which does not necessarily imply causation (they should at least discuss this point). These points must be clarified. Also, the last sub-heading is the most important one, I suggest reducing the first 4 and to expand more the Results on the cophylogenetic analysis. This Figure 4 is quite complicated, and the main message is lost among too many colors and symbols. By expanding the last sub-heading of the Results may help to guide the readers through this part of the Results and to better understand the Figure 4.

The Discussion, in particular the last paragraph, must be improved to make stronger the major claim of the paper stated in the title and in the objective of the study.

Lastly, I think that the title "Horizontal transmission maintains host specificity of symbionts in a brood parasitic host" may be improved to reflect the starting hypothesis of the study.

For an easy navigation through the criticisms please see the area comment in the pdf. The specific comments are also reported in the following list:

Line 1: can be improved

Line 39-42: This is a good start, I recommend to summarize the reasons why the studies are supporting co-diversification as rare event. Also, please clarify that congruent cophylogeny not necessarily means codiversification. Have the author evaluated this point when revising the literature 1 to 9?

Line 42: add incongruence and define as disagreement

Lines 46-47: I suggest to improve:

Several important pieces of literature to cite are missing here and along the Introduction: for example:

<https://doi.org/10.1111/j.1365-2699.2008.01951.x>

<https://doi.org/10.1093/sysbio/syad022>

<https://doi.org/10.1111/j.1461-0248.2008.01237.x>

Line 48: I suggest to split the figure 1 in two figures: the first supporting the introduction and the second linked to the hypothesis of this study. Also, in the Introduction the authors should introduce better the basic concept behind the hypothesis they want to test. See also my comments in the caption in Fig. 1 below.

Line 49: The authors have to properly define this term, see also comment above

Lines 52-54: This is a very interesting point that the authors intended to test with this paper. I suggest to develop more this sentence because is crucial for their hypothesis and analyses of the results.

Lines 62-63: see also a recent meta-analysis

<https://digitalcommons.unl.edu/manter/25/>

Lines 66-68: assuming that any other event is working to generate incongruence between phylogenies...

Lines 75-77:

This is another interesting point (as the comment in green above line 52) that need to be emphasized and better linked to the concept in line 52. In this way this paper will greatly improve the overall clarity for a broader audience of biologists.

Line 84: This reasoning is difficult to follow for researchers not specialized on this biological model.

Specifically, try to clarify:

1- what is the meaning of "case 1"?

1- If *M. bonariensis* is obligate brood parasite, is it "an excellent model" to investigate all the transmission pathways? or just some?

lines 84-85: This part is also critical for the understanding, if you take the perspective of the parasite, how do you distinguish/define non-host specific mites from host-specific mites in the first place?

I strongly suggest to clarify this.

Line 86-87: Is this in the Results section? Please specify.

Should this justification be placed in the M & M section?

Line 87-89: At this point, it should be clear to the reader that *M. bonariensis* is a good model.

However, I think that the justification and the presentation of the hypothesis may be largely improved.

Lines 92-96: All these details can be moved in Methods and this part can be replaced with a paragraph to better develop the comment in 84-85

Line 98-99: This is probably the best point that the author can use to clarify that congruence or incongruence are not generated by just one single processes. The authors mentioned it at the beginning of the Introduction, but later this point went lost where indeed is worth to stress.

lines 103-106: It is unclear to me how the authors support this hypothesis, what are the assumptions supporting this hypothesis? Please try to better link the assumptions provided above (what is already known) with the present hypothesis.

lines 107-108: this is not clear, please clarify

lines 109-111: Please clarify this point by linking it with my comment above (line 84) on the non-host specific mites

line 114: I suggest to merge this subheading with the next one

lines 119-121: Is the citation number 31 clarify the criteria listed by the authors? If not I suggest to better clarify the criteria with citations from the literature. This can be done in Introduction or Methods section.

from line 122: Indeed, I believe that this entire part from line 122 to line 135, does not belong to Result section.

line 130: which classification? the category? if so, the other two are not based mostly on the literature?

lines 137-138: Something missing in this subheading, high prevalence on *M. bonariensis*? higher compared to what? Please improve

line 141: I suggest to report numbers here

line 142: how about the range?

lines 142-143: how about include this list inside the parenthesis above?

line 144: range?

line 148: as above

line 161: what is everything else? please specify. What about mortality?

Also this entire paragraph from line 160 to 163 should go in Methods or at the end of Introduction, as this belongs with the rationale of the study.

lines 163-164: This result is based on a total of 146 specimens collected in Brazil, the author should specify this because the distribution of this species is larger.

lines 164-168: Is this really a result or should be moved in the Discussion instead?

I think it is better to report only and purely results here and lead assumptions based on the results for the discussion section. I suggest to moved most of this paragraph below.

lines 170-174: Assumption does not belong to Results section

line 180: "or" or "and/or" ?

line 180: If I understand well from Methods section, the authors used 69 individuals from 27 species (?) to estimate the number of mites. Why the author did not chose to analyze 149 specimens for each bird species?

line 205: I think that this paragraph is the most important one. I suggest to re-organize the entire Results section and expand the reporting of these results that now are mainly in the supplementary material.

lines 206-209: This belongs to Methods section.

line 207: the gene flow was not introduced in Methods section. Please improve.

lines 216-217: there is a problem here: a host switched?, please check

lines 221-224: Most of it should go in Discussion

line 229: this information should go in the Introduction section

line 249: conspecific?

line 251: of Molothrus-specific mites? specify

line 251: what is the meaning of "compatible" here?

line 264-267: how the authors know that they were forest parents, please clarify

lines 312-316: must be improved to make stronger the major claim of the paper stated in the title and in the objective of the study.

lines 316-318: I think the Results and the presentation of the assumptions and the hypothesis need to be improved to make this point stronger.

line 321: The details in Introduction can be moved here, I would like to see here the justification: why the authors chose 146 bird species different from the model species to test their hypothesis. This will allow to better define the categorization "mites forming exclusive associations" and "non-host specific mites"

line 322: In supplementary material 1 there are 144 unique IDs for Host Specimen, please check it.

line 327: it looks like there are more then 27 species in Supplementary table 2, could you please clarify?

line 380: In the results section the authors reported gene flow information, they should explain here how they calculated it

Table 1: some column headers need to be explained in the caption (see pdf).

Figure 1: This picture maybe more easy to interpret and read if you add:

1- letters for different groups of figures

2- in the scheme on the top, add some text indicating the parental or non-parental relationship, if the reader is not familiar with the definition of vertical and horizontal transmission it takes a while to understand.

3- It seems that the hypotheses are linked to the boxes on the top, are they really?

Figure 1 caption: This hypothesis is not clear to me, I understand the link between conspecific transmission and host specificity but not the link between lower conspecific transmission and higher host switches. The last two are not necessarily linked.

Figure 2: Please improve the caption: from which host species? Spell out the acronyms from the x-axis.

Also in the Figure the y-axis: which host?

Figure 3 color-code: is this a vernacular name to refer to a group of species? or it is based on the categorization above? Please specify

The same for the other names below

Figure 4: Please explain why it is an unknown host

Also: This Figure 4 is quite complicated and the main message is lost among too many colors and symbols. I suggest to simplify it, also by expanding the last sub-heading of the results may also help to guide the readers through this part of the Results

Figure 4 caption: what is GREY ???

Reviewer #3 (Remarks to the Author):

Ectosymbionts that lack an off-host stage are often assumed to be transferred primarily through vertical transmission (parent->offspring) rather than horizontal transmission (between unrelated individuals). Testing this in wild host species is difficult, as there is no simple way to differentiate symbionts that are descendents of vertically transmitted individuals or of horizontally transmitted ones. The authors of this manuscript take advantage of the biology of a brood-parasitic species, the Shiny Cowbird *Molothrus bonariensis*, to provide rare quantitative evidence of the relative importance of vertical vs. horizontal transfer of obligatory ectosymbionts. Like other bird species that do not brood their own young (e.g., megapodes, some cuckoos), *M. bonariensis* have feather mite species that are specific to them that are transferred horizontally through social contact. Brood-parasitic species also have the potential to acquire mites via vertical transmission from their foster parents (various phylogenetically distantly related passerines). The authors found that the cowbirds acquire a smaller proportion of their feather mite fauna from their foster parents than they do via horizontal transmission from conspecific cowbirds after leaving the foster parent's nest. The authors then use a molecular phylogenetic approach to explore the evolutionary origins of three of the feather mite taxa found on *M. bonariensis* (*Amerodectes*, *Proctophyllodes*, and *Trouessartia* spp.). Based on their analyses, past co-speciation and host-switching events, and present-day continued exchange between different host species, all played roles in creating the acarofauna of *M. bonariensis*.

The authors present very convincing data of the importance of horizontal transmission in maintaining host-specific feather mites in this system at ecological time scales. The phylogenetic analysis is a little less convincing, though this may be due in part to my relatively shaky understanding of the statistical underpinnings of BEAST and eMPress. First, although the authors use an impressively large number of feather mite sequences in their analyses (the addition of so many new mite sequences to GenBank is great!), there are a great many missing taxa (as they themselves admit), both for feather mites and for the (unknown) hosts of the missing feather mites. I hope that one of the other reviewers can comment on the effects of missing taxa on estimates of cophylogenetic events such as host-switching vs. cospeciation. Second, mite fossils used to calibrate some of the trees are from taxa that are quite distantly related to feather mites. I expect that this would also influence reconstruction of coevolutionary scenarios.

Although brood-parasitic birds provide a great opportunity to assess vertical vs horizontal transmission of symbionts, the generality of the importance of horizontal transmission for feather mites as a whole needs further investigation in host taxa other than cowbirds. Especially as one might argue that the host-specific mites of brood-parasitic birds are probably the result of strong selection/ecological filtering for mite species with a particular penchant for transferring to other adult bird individuals rather than just to fledglings.

I have uploaded an annotated version of the manuscript pdf that includes my comments as pop-up boxes and text edits. Below I list the more important of them in order of their occurrence in the ms; however, the authors should read through the annotated pdf to see all of my suggestions for changes to syntax, word choice, grammar, spelling and punctuation.

Abstract and throughout: depending on journal formatting style, you may need to write out genus names in full at beginning of sentences rather than using abbreviations.

Introduction, line 45: The highlighted part of this long sentence is rather an awkward construction. I suggest giving the highlighted section its own sentence, in which you elaborate more about what Refs 12-14 are referring to.

Introduction, line 59: This is not true for Epidermoptidae such as *Myialges* spp. that are specialized for dispersal via hippoboscids. This is mentioned later but perhaps you shouldn't imply at this point that no feather mites are adapted for long-range horizontal transmission.

Introduction, lines 65-71: A long run-on sentence! Please break into two sentences for ease of reading.

Introduction, lines 72-73: Maybe vertical transmission can be accurately measured, but such measurements are rare. Please cite publications that include assessing transmission of feather mites from parents to offspring.

Introduction, ~ line 78: Please briefly provide some biological information about *M. bonariensis*, e.g., its geographical distribution, what families of birds it parasitizes, etc.

Introduction, line 78 and elsewhere in ms: Depending on journal formatting guidelines, you may need to provide taxonomic authorities when giving scientific names of species.

Introduction, line 109: Here cite some of the literature that makes these assumptions of dominance of horizontal transmission.

References: see annotated pdf for corrections to some references

Results, line 121: Reference 31 indicates that the sister of *M. bonariensis* is not *M. ater*, but rather the small clade of (*M. aeneus*+*M. ater*).

Results, line 130: "Quill-and-skin mites (QSM) – this classification is based mostly on the literature: these mites are naturally rare (and have low prevalence/abundance)...". Provide references for this statement. Also, do you think this is **truly** low prevalence and abundance for skin and quill-dwelling analogs and pterolichoids? Or just **apparent** rarity due to feather-ruffling methods being a poor way to recover these mites?

Results, line 135: I agree that it's a good idea to separately consider the skin mites that have been documented to transfer horizontally via hippoboscids; however, there is no clear justification for including quill mites together with the skin mites in this category. What is your rationale?

Results, line 220: Why suggest social contact rather than just repeated transmission from foster parents to *M. ater* chicks? Or do you mean that *P. egglestoni* might be transferred between *M. ater* due to social contacts? Either way, it is ambiguous as written.

Discussion, lines 257-259: "In the cowbird system, all *Molothrus*-specific mites were transmitted horizontally via conspecific contacts, which likely occurred gradually with host age as the birds experience an increasing rate of social contact. Feather mites associated with the Australian bushturkeys (Galliformes: Megapodiidae), a host that has minimal parent-to-offspring contact (eggs are buried, young birds are fully fledged on hatching) do display this pattern as well." By saying 'as well', you imply that you have demonstrated gradual accumulation of mite taxa as the *M. bonariensis* aged. But I don't see such data in this manuscript.

Discussion, line 270: Provide a suggestion of how this cowbird-to-foster transfer might occur (e.g., by fighting between female cowbirds and blackbirds when the cowbirds try to lay eggs in host nests?).

Discussion, lines 274-287: This paragraph seems rather poorly organized and internally repetitive. Please organize more logically and concisely.

Discussion, lines 317-318: Please cite some of the references in which this 'traditional view' of the 'sole force' is explicitly stated.

References: see annotated pdf for corrections to some references.

Table 1: Not clear what 'prevalence by mite record' means. Is it 'proportion of all mite specimens'?

Table 2: Most (or maybe all) of the zeros in this table should be 'not applicable' (n/a) instead, e.g., it is impossible for 'alien only' mites to have any representation in the 'Molothrus-specific' column.

We wish to extend gratitude to the reviewers for their comprehensive review of our manuscript. Your efforts are greatly appreciated.

In the following sections, we provide responses and explanations for each of the reviewer's comments. Additionally, we have included the updated Figures 1, 2, and 3 in the final pages of this document.

Once again, thank you for your valuable feedback and contributions.

Reviewers' comments:

Reviewer #1 (Remarks to the Author):

Capitalizing on brood-parasite system, in this study Pedroso et al., investigated the importance of vertical and horizontal transmission on evolutionary dynamics between bird hosts and their feather mite symbionts. I find this study very exciting and well written. The study also includes ample amount of data, and analyses. Also, the figure 1 is very clear, which helps to follow the hypotheses and predictions of this paper.

I only have few minor suggestions/comments for this paper:

R1.Q1. Line 256: This is very interesting. Have you ever tracked diversity/abundance change of mite species during the ontogeny of cowbirds? It would be cool to know if mites from foster parents are dominant in early in life but later in life they get outcompeted by *Molothrus*-specific mites.

R1.A1. An analysis of changes in mite composition through different life stages of the bird host would provide direct evidence of an exclusion-by-competition scenario (=mites from foster parents are outcompeted by *Molothrus*-specific mites in older birds). However, museum samples are not available for all bird life stages, such as nestlings or juveniles. Therefore, additional field sampling would be needed to answer this question. This can be done in the future, but not as part of this study. Our study is based on analyses of mites inhabiting adult birds only and we clearly state this in the Material & Method section.

To clarify this point, we added the following text in the Discussion section:

“It is also possible that mites transmitted from foster parents are replaced by *Molothrus*-specific mites through competitive exclusion. Unfortunately, here we cannot provide direct evidence for this hypothesis because samples of immature bird stages were not available for study”

R1.Q2. Line 323: It is very nice to use the museum skin collections for such a studies. However, did you find comparable mite abundances in the skin collection and wild-captured or roadkill samples. I am somewhat skeptical when it comes to museum skin collections, which are treated heavily for preservation and might not represent realized mite abundances. If there were any differences, please make a disclaimer about it somewhere.

R1.A2. The sampling of museum skins should not confound the results of the study because mites are not likely to detach from their host even after death during regular handling of museum specimens. In the worst-case scenario (which is not generally expected), if for some reason a bird skin is roughly handled in the museum and many mites are dislodged, this would not change the ratio of *Molothrus*-specific to foster-parent mites. This is because the attachment mechanisms of these two mite groups are fundamentally the same. This ratio (but not abundance) is central to the study's main idea. To avoid stochastic effects, low abundance samples were excluded from our analyses.

To make sure that our sampling of museum bird specimens did not bias our result, we recalculated the key statistics using separate museum and field samples. These analyses produced nearly the same results:

Museum samples alone (298 mite records in 108 host specimens):

Horizontal conspecific transmission (qhc): 74.8% (full analysis) vs 74.1% (downsampled)

Vertical/Horizontal interspecific transmission (qvi+ qhi): 18.9% vs 18.4%

Undetermined type of transmission: 6.3% vs 7.3%

Wash+Field Samples alone (67 mite records in 31 host specimens):

Horizontal conspecific transmission (qhc): 74.8% (full analysis) vs 77.6% (downsampled)

Vertical/Horizontal interspecific transmission (qvi+ qhi): 18.9% vs 20.9%

Undetermined type of transmission: 6.3% vs 1.5%

To address the reviewer comment, we added a disclaimer saying "Their realized abundances may be affected by the skin preparation processes in museums, but this generally is not expected to be a substantial factor as mites remain firmly attached to the feathers even after their death" (Supplementary Note 2: Museum Samples). We did not put this disclaimer into the main body of the manuscript because our main results (estimating the transmission rates in different mite groups) are based on presence/absence, not abundance data.

R1.Q3. Line 380: Cophylogenetic analyses: By reading the methods alone I was unable to see if you conducted the cophylogenetic analyses separately for different mite clades (A-D) in the figure 3. Can you provide bit more information on this? Did you run one cophylogenetic analyses or multiple sub cophylogenetic analyses? Also did you do a cophylogenetic analyses on full trees?

R1.A3. Separate cophylogenetic analyses were done because *Molothrus*-associated mites form three phylogenetically independent clades A, B+C, D (as shown by our global phylogenetic analysis of mites associated with passerine birds in Fig. 2 (=old Fig. 3). This is standard practice.

To clarify this point, we modified the "Cophylogenetic analysis" section as follows:

"To quantify the number of different coevolutionary events in the *Molothrus* system, we performed separate parsimony-based reconciliation cophylogenetic analyses in eMPress[®] for each phylogenetically independent mite lineage (Fig. 2): Pterodectinae (*Amerodectes* A), Proctophyllodinae (*Proctophyllodes* B+C), and Trouessartiidae (*Trouessartia* D)."

R1.Q4. Figure 2. Is this figure necessary? For me it is repeating the information in the table 1. But if you decide to keep this figure, can you give colors to each dot based on their species assignment. Then it is easy to see what which two *Molothrus*-specific mite species have the highest abundances.

R1.A4. We removed this figure.

R1.Q5. Figure 3 and 4. I am a colorblind person and it was a bit difficult for me to see the differences in reds and greens. Do you mind changing one of these colours to may be pink? So then everyone can see things better.

R1.A5. We modified Figures 3, 4 to use a color blind-friendly palette as suggested by the reviewer.

R1.Q6. Figure 4. This is a complex figure. I think the figure legend need bit more explanation (a summary) on each of the panels, so it would be easier for readers to follow and understand the figure.

R1.A6. The reviewer requested information that is too extensive to be included in the figure title. Therefore, to account for this comment, we added this information as part of a new subsection 'Host switches and codivergences occur at nearly the same frequencies' in the Results section, and made additional changes to Figure 4 (see R2.Q72) to make it easier for the reader to follow the cophylogenetic scenarios presented in Figure 4.

Reviewer #2 (Remarks to the Author):

This paper evaluates the magnitude of different mechanisms of parasite transfer from one host species to discuss the implications on the patterns of cophylogeny at macroevolutionary scale and to support the claim in the title: Horizontal transmission maintains host specificity of symbionts in a brood parasitic host.

I color-coded my comments in the pdf attached below with green-highlighted (very good points) and yellow-highlighted feedback (critical points that need to be addressed by the authors) to show that although I found very good points raised and interesting insights, I strongly suggest improving all the criticisms raised. Because the topic is interesting for the community of the evolutionary biologists, I encourage the authors to improve the paper and resubmit it after major revisions.

R2.Q1. In general, I suggest improving the presentation of the hypothesis, there are several unclear points, for example the reasoning in lines 82-89 should drive the reader to understand why the H0 was settled as such in line 103, however it is not entirely clear from the perspective of the parasite how the mechanisms of the transfer are linked to the identity of the parasite and the opportunity to switch the host.

R2.A1. The following text was inserted in the Introduction (a brief discussion of mite groups, their host specificity and coevolutionary histories, and transmission modes setting the stage for our hypotheses H0 and H1):

[brood parasitic biology of *Molothrus bonariensis* is explained] “This is a generalist bird parasitizing more than 90 different passerine species in 17 families in South America and beyond⁴¹⁻⁴³. Given that *M. bonariensis* is a brood parasite, one would expect that this bird might have (i) specific, single-host mite species that co-diverged with their hosts over a long evolutionary time and (ii) foster parent mite species whose coevolutionary history has been mostly driven by host shifts rather than co-divergence events. “ [transmission modes are explained]

R2.Q2. Overall, I think that the authors must clarify the meaning of "mites forming exclusive associations" and "non-host specific mites" how do we arrive at this categorization and how it is linked the mechanisms investigated.

R2.A2. It is reasonable to expect that host-specific (single-host) symbionts are likely to co-diverge with their hosts, while multi-host symbionts, due to their nature, will have host shifts as their main co-evolutionary events. This aspect is well explained in the manuscript, so we believe that clarifying the meaning of "exclusive associations" is sufficient to account for this comment. We made the following clarifications (highlighted) in the “Introduction” and “Definition of host specificity categories” sections:

“*Molothrus*-specific mites (i.e., single-host mites consistently found only on *M. bonariensis*) ...” ; “While *Molothrus*-alien mites (i.e., mites found both on *M. bonariensis* and one or more species of its foster parents) ...” ; “...exclusivity of an association with *M. bonariensis* (i.e., a single-host mite species occurring only on *M. bonariensis*)”

R2.Q3. I suggest improving the description of the experimental design which seems flawed by the fallacy of unbalanced taxon sampling (1 species/146 specimens versus 27 species/69 specimens).

R2.A3. Our study analyzed the presence/absence of mite species found on *M. bonariensis*, but not on other birds. We used data from putative foster parent birds (27 species/69 specimens) and exhaustive literature data to identify and compare the mite species found on *Molothrus*. Based on mite identifications it was possible to assign the host specificity categories, i.e., *Molothrus*-specific or foster parent mite species (*Molothrus*-alien). In other words, we did not sample an unbalanced number of taxa, since we analyzed data from only a single taxon, *M. bonariensis*. We clarified this aspect in the Results section as follows:

“To identify *Molothrus* host-specific mites and mites primarily associated with *Molothrus* foster parents, we conducted an extensive mite survey focusing on *M. bonariensis* (144 specimens). To aid in identifying and classifying mites into the *Molothrus*-specific vs *alien* categories, we also sampled mites from its common foster parents (27 species, 69 specimens) (Supplementary Data 1 and 2) and considered the literature data.”

Even though this reviewer’ comment does not apply to our study, just out of curiosity, we randomly downsampled our *Molothrus* data to 69 individuals and obtained very similar results to our full analysis (Table 2):

Horizontal conspecific transmission (qhc): 74.8% (full analysis) vs 72.5% (downsampled)

Vertical/Horizontal interspecific transmission ($q_{vi} + q_{hi}$): 18.9% vs 20.0%
Undetermined type of transmission: 6.3% vs 7.2%

R2.Q4. I also suggest the authors to re-organize the Results subsections. There are 5 subheadings, the first one does not really belong to Results section and can be moved in Methods. The following three sub-headings report different metrics to measure the association between mite species and predefined groups.

R2.A4. As suggested, we moved the majority of text from the first Result subsection to Methods, but left a short summary of the overall sampling scheme to better connect with comment R2.Q3.

R2.Q5. There are several assumptions described in Results section that need to be moved with the presentation of the hypothesis.

R2.A5. We modified the Results section accordingly. For more details, see R2.Q32, R2.Q41, R2.Q43, R2.Q44, and R2.Q47.

R2.Q6. Moreover, I should warn the authors that most of these results is based on associations which does not necessarily imply causation (they should at least discuss this point). These points must be clarified.

R2.A6. In our manuscript, the inference of rates was based on the actual counting of presence and absence data, which is indicative of either presence or absence of host switches depending on the species involved. On a macroevolutionary scale, host-switch or codiversification events were estimated using event-based cophylogenetic methods. We don't think that our results are based on associations/correlations. For host-specific symbionts, host divergence disrupts the parasite gene flow and will drive the parasite divergence (causation). Hosts switches by definition generate incongruent phylogenies in terms of topology and/or divergence dates.

R2.Q7. Also, the last sub-heading is the most important one, I suggest reducing the first 4 and to expand more the Results on the cophylogenetic analysis.

R2.A7. This comment is similar to R1.Q6. We think that the section "**The rate of horizontal social symbiont transmission is high**" is equally as important as the last section "**Host switches and codivergences occur at nearly the same frequencies**". We restructured it better to clarify the transmission rate calculations. See also R2.A47.

R2.Q8. This Figure 4 is quite complicated, and the main message is lost among too many colors and symbols. By expanding the last sub-heading of the Results may help to guide the readers through this part of the Results and to better understand the Figure 4.

R2.A8. This comment is similar to R1.Q6. We re-worked this figure (Fig 3, formerly Fig 4) for clarity and added explanations of co-phylogenetic events in the manuscript text. See also R2.Q72.

R2.Q9. The Discussion, in particular the last paragraph, must be improved to make stronger the major claim of the paper stated in the title and in the objective of the study.

R2.A9. We improved our last paragraph in the Discussion (=Conclusion) as follows (highlighted=new text):

“In summary, our work highlights that symbiont horizontal transmission via conspecific social contacts is an important dispersal mode of *Molothrus*-specific mites onto new host individuals in

the *Molothrus*-feather mite system. This horizontal transmission alone (without parent-offspring vertical transmission) can maintain highly abundant, dominant, and host-specific species of obligate mite symbionts on their hosts. Here, we identified five independently evolved lineages of mites specific to *M. bonariensis* that colonize new generations of hosts exclusively through horizontal route of transmission. These symbionts persist on *M. bonariensis* at high prevalence and abundance despite the constant influx of new diverse mite invaders transmitted vertically from over 90 species of foster parents. On average, mite species dispersing via conspecific horizontal contacts are three times more likely to colonize *M. bonariensis* than mites transmitted vertically via foster parents. Our data therefore provide evidence challenging the traditional view of the importance of vertical transmission as the sole force generating host specificity on a microevolutionary scale. On a macroevolutionary scale, we show that horizontal transmission maintained these *Molothrus*-specific mites on their hosts over a long evolutionary time, at least 1.38 Mya since the split of *Molothrus bonariensis* and *M. ater*. There were both codivergence and host switch events, occurring nearly at the same frequencies. This suggests that macroevolutionary patterns, which are based on rare coevolutionary events cannot be easily generalized from short-term evolutionary trends, such as transmission mode and rates.”

R2.Q10. Lastly, I think that the title “Horizontal transmission maintains host specificity of symbionts in a brood parasitic host” may be improved to reflect the starting hypothesis of the study.

R2.A10. The only aspect of our hypothesis that is not included in the title is codiversification. We therefore modified the title as follows (new text is highlighted):

“Horizontal transmission maintains host specificity and codiversification of symbionts in a brood parasitic host”

For an easy navigation through the criticisms please see the area comment in the pdf. The specific comments are also reported in the following list:

R2.Q11. Line 1: can be improved

R2.A11. Same as R2.Q10 (see above).

R2.Q12. Line 39-42: This is a good start, I recommend to summarize the reasons why the studies are supporting co-diversification as rare event. Also, please clarify that congruent cophylogeny not necessarily means codiversification. Have the author evaluated this point when revising the literature 1 to 9?

R2.A12. It appears that the reviewer questions the results of these studies, perhaps because most of them did not use temporal evidence, so they may overestimate the number of co-divergences. However, this would not change our general statement that co-divergences are rare. We do realize that a congruent cophylogeny does not necessarily mean codiversification, and therefore used double dating of host and symbiont phylogenies (temporal congruence) in our study. We modified this sentence to clarify the importance of both phylogenetic and temporal aspects to assess codivergence:

Old text: “Most studies have suggested that strict codiversification between hosts and their symbionts is rare even for highly host-specific symbionts.”

New text: “The majority of studies have suggested that strict codiversification between hosts and symbionts (i.e., temporal and topological congruence of host and parasite phylogenetic branching pattern) is rare¹⁴, on average, being only 7% of other coevolutionary events¹⁵.”

R2.Q13. Line 42: add incongruence and define as disagreement

R2.A13. Explanation added (highlighted text):

“Generally, cophylogenetic incongruence (i.e., the disagreement between host and symbiont phylogenetic branching patterns at the macroevolutionary scale) may be caused by several evolutionary events, such as duplication (speciation of a symbiont within a single host species), sorting (extinction and missing the boat), failure of the symbiont to speciate, and host switching (or host shift)”

R2.Q14. Lines 46-47: I suggest to improve:

Several important pieces of literature to cite are missing here and along the Introduction: for example:

<https://doi.org/10.1111/j.1365-2699.2008.01951.x>

<https://doi.org/10.1093/sysbio/syad022>

<https://doi.org/10.1111/j.1461-0248.2008.01237.x>

R2.A14. We added these references and updated our text as suggested by the reviewer (added text is highlighted): “At the microevolutionary scale, host switching is also a biologically intriguing event leading to the evolution of multihost symbionts, especially when it occurs between phylogenetically distant hosts^{1,18,19}”

R2.Q15. Line 48: I suggest to split the figure 1 in two figures: the first supporting the introduction and the second linked to the hypothesis of this study. Also, in the Introduction the authors should introduce better the basic concept behind the hypothesis they want to test. See also my comments in the caption in Fig. 1 below [R2.Q67].

R2.A15. We split figure 1 into two figures A and B, and kept them in the same figure plate because of their interdependence.

The “basic concepts” comment was addressed in R2.A24 and R2.A27 (Introduction).

For the Fig. 1 comment, see R2.A67.

R2.Q16. Line 49: The authors have to properly define this term, see also comment above

R2.A16. Changed as suggested, see R2.A13.

R2.Q17. Lines 52-54: This is a very interesting point that the authors intended to test with this paper. I suggest to develop more this sentence because it is crucial for their hypothesis and analyses of the results.

R2.A17.

L52-54: “Yet, despite perceived dominance of vertical transmission, some host-symbiont systems may simultaneously display both incongruent cophylogenetic patterns and high host specificity^{4,6,10,19,20}.”

We modified this sentence as follows (highlights): “Yet, despite the perceived dominance of vertical transmission and low horizontal transmission rates²⁴⁻²⁸, some host-symbiont systems may simultaneously display both incongruent cophylogenetic patterns and high host specificity^{7,9,16,29,30}.”

This conundrum challenges the role of vertical conspecific transmission in promoting codiversification and maintaining host specificity”.

R2. Q18. Lines 62-63: see also a recent meta-analysis
<https://digitalcommons.unl.edu/manter/25/>

R2.A18. These lines discuss expectations of a predominant co-evolutionary pattern in feather mites. However, the meta-analysis suggested by the reviewer is based on all host-symbiont systems, so it may not be appropriate here. We included this reference in the Introduction section.

R2.Q19. Lines 66-68: assuming that any other event is working to generate incongruence between phylogenies...

R2.A19.

Old text L66-68: “Given the discordance of current biologically informed expectations (i.e., vertical transmission should be common, hence cophylogenies should be congruent vs. observations of widespread cophylogenetic incongruence while mites maintain high host specificity), it is important to understand the relative contribution of different transmission modes into these associations. “

We entirely re-wrote this sentence (highlights indicate “other” events) : “... Consequently, the diversification of these symbionts is expected to be driven largely by host diversification. However, multiple cophylogenetic studies have shown that host switches are relatively common in feather mites^{7,9,10,32,40}, suggesting that host switches are in fact one of the main drivers of feather mite diversification⁷. Thus, the current biological expectations are in conflict: one suggests that vertical transmission should be prevalent, leading to congruence between host and symbiont phylogenies, while observations show widespread phylogenetic incongruence among mites, despite their high host specificity. Therefore, it is crucial to understand the relative contribution of two types of conspecific transmission: vertical and horizontal (q_{vc} and q_{hc}), promoting cophylogenetic concordance and high host specificity vs interspecific transmission (q_{vi} and q_{hi}), that can generate cophylogenetic discordance and low host specificity (Fig. 1).”.

R2.Q20. Lines 75-77: This is another interesting point (as the comment in green above line 52) that need to be emphasized and better linked to the concept in line 52. In this way this paper will greatly improve the overall clarity for a broader audience of biologists.

R2.A20. For lines 75-77, we extended this sentence to emphasize and link it with our hypothesis and our study system as follows (For L52, see also R2.Q17):

Old text L52, green: “Yet, despite perceived dominance of vertical transmission, some host-symbiont systems may simultaneously display both incongruent cophylogenetic patterns and high host specificity^{4,6,10,19,20}.”)

Old text L75-77: “As a result, vertical transmission (q_{vc}) has been overemphasized while conspecific horizontal transmission (q_{hc}) has been largely overlooked in the literature.”

New text L52, green: “Yet, despite perceived dominance of vertical transmission **in the literature**, some host-symbiont systems may simultaneously display both incongruent cophylogenetic patterns and high host specificity^{4,6,10,19,20}.”

New text L75-77: “As a result, vertical transmission (q_v) has been overemphasized while conspecific horizontal transmission (q_{hc}) has been largely overlooked in the literature. This has contributed to the uncertainty regarding the role of conspecific horizontal transmission in shaping both host specificity and cophylogenetic congruence.”

R2.Q21. Line 84: This reasoning is difficult to follow for researchers not specialized on this biological model. Specifically, try to clarify:

1- what is the meaning of "case 1"?

R2.A21.1. The confusing referencing (case 1, case 2) were removed throughout the manuscript and Fig. 1 was redone (see R2.A15 above).

2- If *M. bonariensis* is obligate brood parasite, is it "an excellent model" to investigate all the transmission pathways? or just some?

R2.A21.2. We added the following clarification (highlighted):

“The brood parasitic shiny cowbird, *Molothrus bonariensis* (Passeriformes: Icteridae), provides an excellent model to investigate the relationship between conspecific and interspecific transmission and host specificity in feather mites.”

R2.Q22. lines 84-85: This part is also critical for the understanding, if you take the perspective of the parasite, how do you distinguish/define non-host specific mites from host-specific mites in the first place? I strongly suggest to clarify this.

R2.A22 Host-specific=one host (*Molothrus*), non-host specific=found on more than one host (*Molothrus* and one or more species of its foster parents). From a parasite perspective this translates into “I want to stay on a single host species, *Molothrus*” (host-specific) vs “I want to parasitise multiple host species, *Molothrus* and its foster parents” (non-host specific). There could be various factors contributing to why certain symbionts exhibit host-specificity while others do not. This is well described in the literature, and this “parasite perspective” is outside the scope of our manuscript.

We re-wrote the entire text to make it clear.

Old text L82-85: “... mites forming exclusive associations with *M. bonariensis* (host-specific mites) can be transferred only by horizontal contact between conspecific hosts, i.e., other *M. bonariensis* (Fig. 1: case 1: q_{hc}), whereas non-host specific mites (i.e., from *M. bonariensis* foster parents) would be transferred mostly via vertical interspecific care (Fig. 1: case 1: q_{vi})”.

New text: “*Molothrus*-specific mites (i.e., single-host mites consistently found only on *M. bonariensis*) can be transferred only by horizontal contact between conspecific hosts, i.e., other *M. bonariensis* (Fig. 1A: q_{hc}). While *Molothrus*-alien mites (i.e., mites found both on *M. bonariensis* and one or more species of its foster parents) would be transferred mostly via vertical interspecific care (Fig. 1A: q_v) ...”

R2.Q23. Line 86-87: Is this in the Results section? Please specify. Should this justification be placed in the M & M section?

R2.A.23.

The original text: “.. (Fig. 1A: q_{hi} ; see justification in the section "*Molothrus-specific* mites have higher species richness ...”).”

We agree that this phrase refers to our results. However, we would like to keep it in the Introduction, because otherwise our reasoning will not be clear.

R2.Q24. Line 87-89: At this point, it should be clear to the reader that *M. bonariensis* is a good model. However, I think that the justification and the presentation of the hypothesis may be largely improved.

R2.A24. We added the justification for our hypothesis as follows:

“At the macroevolutionary timescale, the constant transmission of mites from foster parents likely has provided more opportunities for host switches than what would have been expected in non-brood parasite bird-mite systems. The *M. bonariensis* system, therefore, can be used to evaluate the magnitude of horizontal conspecific transmission (q_{hc}) vs. interspecific transmissions (i.e., vertical q_{vi} and horizontal q_{hi}) and their influence in long-term co-evolution in host-symbiont systems.”

R2.Q25. Lines 92-96: All these details can be moved in Methods and this part can be replaced with a paragraph to better develop the comment in 84-85

Original L92-96: “To identify *Molothrus* host-specific mites and mites associated with *Molothrus* foster parents, we conducted an extensive mite survey from specimens of *M. bonariensis* and 27 species (69 individuals) of foster parents, and analyzed patterns of their host specificity and evolution using various lines of evidence: exclusivity of an association, genetic distances between mites on different hosts, and dated phylogenetics of mites.”

R2.A25. As suggested, we incorporated lines 92-96 into the Methods and Results sections. See R2.A22 above for changes relevant to L84-85.

R2.Q26. Line 98-99: This is probably the best point that the author can use to clarify that congruence or incongruence are not generated by just one single processes. The authors mentioned it at the beginning of the Introduction, but later this point went lost where indeed is worth to stress.

R2.A26. Changed as suggested:

Old text (L98-99): “Finally, we used cophylogenetic reconciliation analyses to quantify the number of coevolutionary events that occurred on the macroevolutionary scale in this system.”

New text: “Then we used event-based cophylogenetic reconciliation analyses to estimate the number of four coevolutionary events (codivergence, duplication, host switch, extinction) that occurred in this system on the macroevolutionary scale.”

R2.Q27. lines 103-106: It is unclear to me how the authors support this hypothesis, what are the assumptions supporting this hypothesis? Please try to better link the assumptions provided above (what is already known) with the present hypothesis.

R2.A27. We modified this text as follows (new text is highlighted):

“Our null hypothesis is that if horizontal conspecific transmission (q_{hc}) is lower than interspecific transmission (rates q_{vi} and q_{hi}) in the *Molothrus*-system, then both host specificity and cophylogenetic congruence should be low due to a high frequency of interspecific transmission from foster parents (q_{vi}) and the absence of conspecific vertical transmission (q_{vc}) (Fig. 1B: H_0).”

R2.Q28. lines 107-108: this is not clear, please clarify

R2.A28. In this phrase we meant that the rate of host switch events cannot be predicted as they may have occurred independently from transmission rates because many host switch opportunities may arise over a long evolutionary time.

We modified the text as follows (L107-108):“ ... host specificity should be high, while the level of cophylogenetic congruence cannot be predicted because many host switch opportunities may arise over a long evolutionary time.”

R2.Q29. lines 109-111: Please clarify this point by linking it with my comment above (line 84) on the non-host specific mites

R2.A29. We have clarified our definition of “non-specific mites” mentioned in line 84

(R2.Q22). Lines 109-111 meant to clarify that using the *Molothrus* system, we can test whether host-specific symbionts can persist via horizontal transmission route only. For changes, see R2.A9.

R2.Q30. line 114: I suggest to merge this subheading with the next one

R2.A30. Subheadings merged as follows: “*Molothrus-specific* mites have higher species richness, prevalence, and abundance than *Molothrus-alien* mites”

R2.Q31. lines 119-121: Is the citation number 31 clarify the criteria listed by the authors? If not I suggest to better clarify the criteria with citations from the literature. This can be done in Introduction or Methods section.

R2A31. The relevant text (L117-121) is: “we identified mite species and assigned them in three host specificity categories (*Molothrus-specific*, *Molothrus-alien* and *Quill-and-skin (QSM)* mites) based on the following criteria: exclusivity of an association with *M. bonariensis*, phylogenetic relationships and genetic distances as compared to mites from foster parents or one of its sister brood parasitic species, *Molothrus ater*”³¹”

Ref 31 does not define any of these criteria, it only reports that *M. bonariensis* and *M. ater* are sister clades. We define these criteria in the present work and provide a reference (Wells and Clark, 2019), see the Methods section and Supplementary Note 2.

R2.Q32. from line 122: Indeed, I believe that this entire part from line 122 to line 135, does not belong to Result section.

R2.A32. We moved the majority of this text to the Methods section, but still kept a short text here for context: “Mites were identified and assigned to three categories, *Molothrus-specific*, *Molothrus-alien*, and quill-and-skin mites QSM; with the former two categories representing feather vane mites lacking a vector transmission, while the latter category representing rare ecological groupings, some of which can be transmitted by a vector (see Methods).”

R2.Q33. line 130: which classification? the category? if so, the other two are not based mostly on the literature?

R2.A33. Line 130 describes the QSM category. We completely re-wrote this paragraph addressing this comment and also R3.Q14

Old text: “*Quill-and-skin* mites (*QSM*) – this classification is based mostly on the literature: these mites are naturally rare (and have low prevalence/abundance), live on the host skin (e.g., *Microlichus*, *Metamicrolichus*, *Strelkoviacarus*) and inside quills (*Dermoglyphus*). In contrast, mites in the two former categories inhabit vanes or down feathers. The skin-inhabiting mites have the ability to be horizontally transferred across interspecific and intraspecific hosts via phoresy on louse flies (Diptera: Hippoboscidae).³²⁻³⁴”

New text: “*Quill-and-skin* mites (*QSM*) – inhabit the host skin (e.g., *Microlichus*, *Metamicrolichus*, *Strelkoviacarus*) or live inside feather quills (*Dermoglyphus*). These mites are grouped together because of their restricted contact transmission in comparison to that of plumage feather mites (groups i and ii). For example, when two bird individuals come into contact, the likelihood of sharing skin or quill mites is significantly lower as compared to that of plumage mites. Furthermore, skin mites are naturally rare and have low prevalence and abundance⁷² in comparison with plumage mites⁷³. Some skin-inhabiting mites also have the ability to be horizontally transferred across interspecific and intraspecific hosts via phoresy on louse flies (Diptera: Hippoboscidae)^{36,74,75}. As a result, in terms of their transmission patterns, these mites cannot be directly compared to plumage-inhabiting feather mites (groups i and ii).”

R2.Q34. lines 137-138: Something missing in this subheading, high prevalence on *M. bonariensis*? higher compared to what? Please improve

R2.A34. This subheading title was modified as follows: “*Molothrus-specific* mites have higher species richness, prevalence, and abundance than *Molothrus-alien* mites”

R2.Q35. line 141: I suggest to report numbers here

R2.A35. We replaced “multiple” by the exact number (130), as requested:

Old text: “*Molothrus-specific* mites (5 species) were exclusively and consistently found on multiple *M. bonariensis* specimens with moderate to high prevalence”

New text: “*Molothrus-specific* mites (5 species) were exclusively and consistently found on 130 *M. bonariensis* specimens with moderate to high prevalence”

R2.Q36. line 142: how about the range?

R2.A36. Changed as suggested:

Old text: “(>13%, Table 1)”

New text: “(13-60%, Table 1:”

R2.Q37. lines 142-143: how about include this list inside the parenthesis above?

R2.A37. We included the list inside parentheses as follows:

Old text: “*Molothrus-specific* mites (5 species) were exclusively and consistently found on multiple *M. bonariensis* specimens with moderate to high prevalence (>13%, Table 1): *Proctophyllodes molothrus*; *Amerodectes molothrus*; *Xolalgoides* sp.1; *Mesalgoides* sp.1; and *Trouessartia* sp.6^{35,36}.”

New text: “*Molothrus-specific* mites (5 species) were exclusively and consistently found on 130 *M. bonariensis* specimens with moderate to high prevalence (13-60%, Table 1: *Proctophyllodes molothrus*, *Amerodectes molothrus*, *Xolalgoides* sp.1, *Mesalgoides* sp.1, and *Trouessartia* sp.6^{44,45}).”

R2.Q38. line 144: range?

R2.A38. Changed as suggested:

Old text: “(prevalence < 7.5%)”

New text: “(prevalence 0.69-7.5%)”

R2.Q39. line 148: as above

R2.A39. Changed as suggested:

Old text: “(< 5%)”

New text: “(1.4-5.6%)”

R2.Q40. line 161: what is everything else? please specify. What about mortality?

R2.A40. We re-wrote this sentence to avoid using “everything else” and then moved it to the Introduction section (see below). We did not estimate mite mortality.

Old text L161-163: “Assuming that each mite species on each *M. bonariensis* individual resulted from at least a single colonization event, and everything else being equal, the distribution of mites in each host specificity category should reflect the relative conspecific and interspecific transmission rates (Table 2).”

New text: “Specifically, we evaluate the effective horizontal versus vertical mite transmission rates assuming that each mite species on each *M. bonariensis* specimen resulted from at least a single successful host switch event. The distribution of mites in each host specificity category, therefore, should reflect the relative conspecific and interspecific transmission rates.”

R2.Q41. Also this entire paragraph from line 160 to 163 should go in Methods or at the end of Introduction, as this belongs with the rationale of the study.

R2.A41. We moved this paragraph to the Introduction section. For details, see R2.A40 above.

R2.Q42. lines 163-164: This result is based on a total of 146 specimens collected in Brazil, the author should specify this because the distribution of this species is larger.

R2.A42. Our Materials section already specifies this: “We sampled 144 *M. bonariensis* specimens for feather mites in Brazil ...”.

R2.Q43. lines 164-168: Is this really a result or should be moved in the Discussion instead?
I think it is better to report only and purely results here and lead assumptions based on the results for the discussion section. I suggest to moved most of this paragraph below.

R2.A43. We believe that these lines, although giving some explanations for clarity, report only the results obtained in our study. Thus, these lines do belong to the Result section. We re-wrote this sentence, so it does not sound like a discussion.

Old text: "... Since *Molothrus*-specific mites can only disperse between *M. bonariensis* via horizontal conspecific transmission (q_{hc}) and *Molothrus*-alien mites predominantly disperse via vertical interspecific transmission from foster parents (q_{vi}), **one can conclude** that cowbird-to-cowbird transmission is at least 3.9 times more frequent than foster parent transmission in this system."

New text: "... Since *Molothrus-specific* mites can only disperse between *M. bonariensis* via horizontal conspecific transmission (q_{hc}) and *Molothrus-alien* mites predominantly disperse via vertical interspecific transmission from foster parents (q_{vi}), **it can be inferred** that cowbird-to-cowbird transmission is at least 3.9 times more frequent than foster parent transmission in this system. "

R2.Q44. lines 170-174: Assumption does not belong to Results section.

R2.A44. We respectfully disagree. The assumption clause gives context clarifying how we arrived at this result. However, we-slightly re-phrase this text for cklarity (highlights)

Old text: "Assuming that QSM mites disperse either only via (foster) parental care or via host horizontal contacts, we estimate the overall ratio ..."

New Text: Assuming that QSM mites disperse either only via (foster) parental care or **only** via host horizontal contacts, we estimate the overall ratio ...

R2.Q45. line 180: "or" or "and/or" ?

R2.A45. Changed as suggested, "or" -> "and/or"

R2.Q46. line 180: If I understand well from Methods section, the authors used 69 individuals form 27 species (?) to estimate the number of mites. Why the author did not chose to analyze 149 specimens for each bird species?

R2.A46. Because we estimated the number of mite species in the *Molothrus*-system only. Foster parents were examined to validate our host specificity categories. To determine that a mite species is *alien* it should be found in at least one foster species species, so there is no need for extra-deep sampling of each foster parent species. This comment is the same as R2.Q3 (see our detailed response in R2.A3 above).

R2.Q47. line 205: I think that this paragraph is the most important one. I suggest to re-organize the entire Results section and expand the reporting of these results that now are mainly in the supplementary material.

R2.A47. L205 is a subheading titled "Host switches and codivergences occur at nearly the same frequencies". We merged this subsection with "Supplementary Note 5" which belongs here. This comment is the same as R2.Q7 (see above).

R2.Q48. lines 206-209: This belongs to Methods section.

R2.A48. We believe that giving a brief, one-sentence introduction of our methods would be beneficial for understanding our results. This is a common practice in papers published in the Nature group journals. We, however, modified this sentence as follows:

Old text: “We performed cophylogenetic analyses on dated host and parasite phylogenies (Fig. 4, Supplementary Note 4, Supplementary Figs. 1-5) to identify either ongoing (with gene flow) or historical host switching (no gene flow) of aforementioned mites from *Molothrus* and their close relatives.”

New text: “For the three mite lineages, A, B+C, and D (Fig. 3), we performed separate cophylogenetic analyses using dated host and parasite phylogenies (Fig. 3, Supplementary Note 4, Supplementary Figs. 1-5) to identify either ongoing (with gene flow) or historical host switching (no gene flow).”

R2.Q49. line 207: the gene flow was not introduced in Methods section. Please improve.

R2.A49. An explanation was added: “Ongoing gene flow was considered to be present between two mite populations from different hosts if their CO1 K2P distances were equal or lower than 5%^{76,86}.”

R2.Q50. lines 216-217: there is a problem here: a host switched?, please check

R2.A50. Typo corrected.

Old text “However, host switches occurred in their ancestral nodes: in *Amerodectes*, a host switched from ancestral *Molothrus* to ancestral *Sicalis* (Fig. 4A)”.

New text: “However, host switches occurred in their ancestral nodes: in *Amerodectes*, there was a host switch from ancestral *Molothrus* to ancestral *Sicalis* (Fig. 3A)”.

R2.Q51. lines 221-224: Most of it should go in Discussion

R2.A51. As suggested, we moved lines 221-224 to the Discussion section with a minor clarification (highlighted) :

Old text: “The coevolutionary patterns of *Molothrus*-associated mites exhibit a gradient of outcomes, from strict co-divergence to ongoing gene flow and complete host switches, providing snapshots of how low rate of interspecific transmission over macroevolutionary time scale could result in potential host switch events.”

New text: “The coevolutionary patterns of *Molothrus*-associated mites, however, exhibit a range of possible events, from strict codivergence to incomplete and complete host switches, providing snapshots of how a low rate of interspecific transmission over macroevolutionary time scale could result in potential host switch events.”

R2.Q52. line 229: this information should go in the Introduction section

R2.A52. The relevant text is L229 “... we studied a generalist brood parasitic passerine ...”. With all due respect to the reviewer, we think it is a good idea to remind the reader that this bird is a brood parasite. The Introduction section already states this.

R2.Q53. line 249: conspecific?

R2.A53. We added the word “conspecific” in this sentence.

R2.Q54. line 251: of *Molothrus*-specific mites? Specify

R2.Q55. line 251: what is the meaning of "compatible" here?

R2.A54. We modified this sentence for clarity by adding explanations (highlights), “compatible” was changed to “equal”: “Under an alternative scenario, when horizontal interspecific transmission of *Molothrus*-alien mites is equal or greater than horizontal conspecific transmission of *Molothrus*-specific mites (i.e., $q_{hi} \geq q_{hc}$), hosts harboring alien-only mites would be expected to be found at the same frequency or greater as hosts harboring only *Molothrus*-specific mites. However, this is not the case in this system in Brazil.”

R2.Q56. line 264-267: how the authors know that they were forest parents, please clarify

R2.A56. *Amerodectes molothrus* (lineage A): These mites switched from *Molothrus* to *Sicalis*; the birds of the genus *Sicalis* are known to be one of the very common foster parents of *Molothrus*. *Proctophyllodes molothrus* (lineage B): These mites had a host switch to *Molothrus* from an unknown host; unfortunately, we do not know whether this host was a foster parent or not. For the purpose of this sentence (it just gives examples of host switches) it is not necessary to specify whether the source or target hosts were foster parents. We modified the text so it is accurate:

Old text L264-267: “As an example, the two most common and abundant mite species, *Amerodectes molothrus* and *Proctophyllodes molothrus*, were involved in historical host switches from foster parents (*P. molothrus*) or in the opposite direction (*A. molothrus*).”

New text: “As an example, the two most common and abundant mite species, *Amerodectes molothrus* and *Proctophyllodes molothrus*, were involved in historical host switches (Fig. 3A, B).”

R2.Q57. lines 312-316: must be improved to make stronger the major claim of the paper stated in the title and in the objective of the study.

R2.A57. See R2.Q9.

R2.Q58. lines 316-318: I think the Results and the presentation of the assumptions and the hypothesis need to be improved to make this point stronger.

R2.A58. See R2.Q9.

R2.Q59. line 321: The details in Introduction can be moved here, I would like to see here the justification: why the authors chose 146 bird species different from the model species to test their hypothesis. This will allow to better define the categorization "mites forming exclusive associations" and "non-host specific mites"

R2.A59. This is a misunderstanding. L321 actually says “We sampled 146 shiny cowbirds specimens for feather mites in Brazil, ...”

We actually sampled specimens (not species as the reviewer suggests), and we sampled the shiny cowbird (*Molothrus bonariensis*) which is our model species (the reviewer says “different from the model species”).

A large sampling of *Molothrus* was needed because we used these data to estimate the transmission rates. We did not estimate transmission rates in foster parent mite species, so the sampling is less intense here. We answered the same comments about our sampling in R2.Q3 and R2.Q46 and also assessed the sensitivity of our full analysis using a downsampled dataset (see R2.A3).

To avoid any misunderstanding we added clarifications as follows: “... This dataset was used to estimate the mite transmission rates ... To identify foster parent mite species and classify mites into *Molothrus-specific* and *Molothrus-alien* categories, we also sampled mites

R2.Q60. line 322: In supplementary material 1 there are 144 unique IDs for Host Specimen, please check it.

R2.A60. We indeed had 144 specimens. We recalculated all our analyses and updated the text accordingly. We do thank the reviewer for this comment.

R2.Q61. line 327: it looks like there are more than 27 species in Supplementary table 2, could you please clarify?

R2.A61. Supplementary Data 2 lists mite specimens used for molecular work. To facilitate data representation and retrieval, we added a new column titled “Data Description” with the following categories: putative foster parent (this study), *Molothrus bonariensis* (this study), *Molothrus ater* (this study), GenBank. The “putative foster parent” category lists mites and their hosts relevant to this comment. To avoid ambiguities, text was changed as follows:

Old text: “We also sampled mites from another 27 passerine species (69 specimens, 21 genera, 10 families) captured with mist nets from different regions in Brazil for molecular and morphological assessment (Supplementary Data 2).”

New text: “... we also sampled mites from (i) putative *Molothrus bonariensis* foster parent species⁴³: 68 specimens (27 species, 19 genera, 10 families) captured with mist nets from different regions in Brazil, and (ii) 1 specimen of *Molothrus ater*, which is sister to *M. bonariensis*⁷¹, from Mexico (Supplementary Data 2: column “Data description”).

R2.Q62. line 380: In the results section the authors reported gene flow information, they should explain here how they calculated it

R2.A62. Same as R2.Q49, see R2.A49 above.

R2.Q63. Table 1: some column headers need to be explained in the caption (see pdf).

R2.A63. Explanations, added, see Table 1 header in the manuscript.

Figure 1: This picture maybe more easy to interpret and read if you add:

R2.Q64. 1- letters for different groups of figures

R2.A64. Letters added: A (for different transmission modes and systems), B (for the two hypotheses). See also R2.Q15.

R2.Q65. 2- in the scheme on the top, add some text indicating the parental or non-parental relationship, if the reader is not familiar with the definition of vertical and horizontal transmission it takes a while to understand.

R2.A65. We divided section A into two groups: parental transmissions and non-parental transmission.

R2.Q66. 3- It seems that the hypotheses are linked to the boxes on the top, are they really?

R2.A66. The box on top is a visual representation of our system. It helps the reader to understand the different transmission modes in the different bird systems. We re-arranged Fig. 1 (see R2.Q15), and believe that the aspect discussed in the comment is clear.

R2.Q67. Figure 1 caption: This hypothesis is not clear to me, I understand the link between conspecific transmission and host specificity but not the link between lower conspecific transmission and higher host switches. The last two are not necessarily linked.

R2.A67. In our study, we compare the rates of conspecific vs. interspecific transmissions in only two relevant scenarios (H_0 and H_1). H_0 considers a scenario of lower conspecific transmission and higher interspecific transmission (=multi-host symbionts are more prevalent than host-specific symbionts). As the reviewer suggests, testing a link between lower conspecific transmission alone and host switches is meaningless, and we do not consider such a hypothesis. To account for this comment, we added the following clarification (highlighted):

“... Our null hypothesis (H_0) proposes that a lower conspecific transmission rate (in comparison to the interspecific rate) results in lower host specificity, lower number of codivergence events, and higher number of host switches ...”

R2.Q68. Figure 2: Please improve the caption: from which host species? Spell out the acronyms from the x-axis.

R2.A68. We removed this figure as per R1.Q4.

R2.Q69. Also in the Figure the y-axis: which host?

R2.A69. We removed this figure as per R1.Q4.

R2.Q70. Figure 3 color-code: is this a vernacular name to refer to a group of species? or it is based on the categorization above? Please specify. The same for the other names below

R2.A70. The color-code legend was modified to avoid ambiguities. For example, “*Molothrus bonariensis* mites” was changed to “Mites associated with *Molothrus bonariensis*”.

R2.Q71. Figure 4: Please explain why it is an unknown host

R2.A71. Fig. 3B (formerly Fig 4B) shows a cophylogenetic scenario for the existing mite lineage is the *Proctophyllodes molothrus* species group (group B). Based on our molecular phylogenetic analysis (Fig. 2), this group is sister to a mite lineage having hosts from many families (Cardinalidae, Icteriidae, Passerellidae, and Icteridae). Given this uncertainty, the ancestral host of group B cannot be determined. We added the following explanation to the manuscript text: “Because this group is sister to the mite lineage having hosts from many families (Cardinalidae, Icteriidae, Passerellidae, and Icteridae) (Fig. 2B), its ancestral host cannot be identified with certainty.”

R2.Q72. Also: This Figure 4 is quite complicated and the main message is lost among too many colors and symbols. I suggest to simplify it, also by expanding the last sub-heading of the results may also help to guide the readers through this part of the Results.

R2.A72. We streamlined Fig. 3 (formerly Fig. 4) and expanded the subsection “Host switches and codivergences ...“ of the Results section to include detailed descriptions of co-evolutionary scenarios A-D from Supplementary Note 5. See the manuscript text.

R2.Q73. Figure 4 caption: what is GREY???

R2.A73. Grey was removed to avoid confusion and Fig 3 (formerly Fig. 4) was reworked, see the previous comment.

Reviewer #3 (Remarks to the Author):

Ectosymbionts that lack an off-host stage are often assumed to be transferred primarily through vertical transmission (parent->offspring) rather than horizontal transmission (between unrelated individuals). Testing this in wild host species is difficult, as there is no simple way to differentiate symbionts that are descendents of vertically transmitted individuals or of horizontally transmitted ones. The authors of this manuscript take advantage of the biology of a brood-parasitic species, the Shiny Cowbird *Molothrus bonariensis*, to provide rare quantitative evidence of the relative importance of vertical vs. horizontal transfer of obligatory ectosymbionts. Like other bird species that do not brood their own young (e.g., megapodes, some cuckoos), *M. bonariensis* have feather mite species that are specific to them that are transferred horizontally through social contact. Brood-parasitic species also have the potential to acquire mites via vertical transmission from their foster parents (various phylogenetically distantly related passerines). The authors found that the cowbirds acquire a smaller proportion of their feather mite fauna from their foster parents than they do via horizontal transmission from conspecific cowbirds after leaving the foster parent's nest. The authors then use a molecular phylogenetic approach to explore the evolutionary origins of three of the feather mite taxa found on *M. bonariensis* (*Amerodectes*, *Proctophyllodes*, and *Trouessartia* spp.). Based on their analyses, past co-speciation and host-switching events, and present-day continued exchange between different host species, all played roles in creating the acarofauna of *M. bonariensis*.

The authors present very convincing data of the importance of horizontal transmission in maintaining host-specific feather mites in this system at ecological time scales. The phylogenetic analysis is a little less convincing, though this may be due in part to my relatively shaky understanding of the statistical underpinnings of BEAST and eMPress.

R3.Q1. First, although the authors use an impressively large number of feather mite sequences in their analyses (the addition of so many new mite sequences to GenBank is great!), there are a great many missing taxa (as they themselves admit), both for feather mites and for the (unknown) hosts of the missing feather mites. I hope that one of the other reviewers can comment on the effects of missing taxa on estimates of cophylogenetic events such as host-switching vs. cospeciation. Second, mite fossils used to calibrate some of the trees are from taxa

that are quite distantly related to feather mites. I expect that this would also influence reconstruction of coevolutionary scenarios.

R3.A1. Incomplete phylogenies may overestimate codivergencies when related non-sister host taxa are inferred as sisters due to missing taxa. However, in the key portion of our phylogeny (*Molothrus*), there were no missing taxa because we sampled the true sister species, *Molothrus bonariensis* and *Molothrus ater*. So, co-divergence events are not expected to be overestimated. The presence of missing taxa may also underestimate extinction events (losses) and overestimate the failure-to-speciate event, but our work mostly focuses on host switches vs co-divergences.

Fossil calibration points of feather mites were not included because there are no fossil feather mites. There is a claim of fossil eggs of “feather mites” in the literature [1], but these are actually not feather mites [2]. Feather mites have elongated eggs, but eggs described in reference [1] are spherical.

Despite using distant outgroups for time-calibration, our mite divergence time estimates are generally coherent and are supported through independent validation by temporally coinciding intercontinental bird-mite co-dispersals followed by a diversification in the new geographical regions. A more detailed discussion can be found in reference [3].

1. Martill, D. M., & Davis, P. G. (1998). Did dinosaurs come up to scratch? *Nature*, 396(6711), 528-529. doi:10.1038/25027
2. Proctor, H. C. (2003). Feather mites (Acari: Astigmata): ecology, behavior and evolution. *Annual Review of Entomology*, 48, 185-209.
3. Klimov, P. B., Mironov, S. V., & OConnor, B. M. (2017). Detecting ancient codispersals and host shifts by double dating of host and parasite phylogenies: Application in proctophyllodid feather mites associated with passerine birds. *Evolution*, 71(10), 2381-2397. doi:10.1111/evo.13309

R3.Q2. Although brood-parasitic birds provide a great opportunity to assess vertical vs horizontal transmission of symbionts, the generality of the importance of horizontal transmission for feather mites as a whole needs further investigation in host taxa other than cowbirds. Especially as one might argue that the host-specific mites of brood-parasitic birds are probably the result of strong selection/ecological filtering for mite species with a particular penchant for transferring to other adult bird individuals rather than just to fledglings.

R3.Q2. “Regular” mites (associated with birds with parental care) may also have a strong selection for transferring to adult birds. This is because adult birds have a higher probability to survive as compared to fledglings. Adults birds can interact with other unrelated conspecific birds and mate, so mites, if they stay on adult hosts, also have a higher probability to survive and colonize new host individuals. Finally, there are studies supporting that “regular” mites do prefer adult birds over fledglings⁶⁹⁻⁷¹ (see below). Co account for the reviewer comment, we added a cautious statement saying our finding may represent a general pattern as follows:

“Even in non-brood parasitic systems, horizontal conspecific transmission has also been observed as the main transmission route of symbionts, such as feather mites from red-billed choughs⁶⁴, barn swallows⁶⁵, and in feather lice from bee-eaters⁶⁴⁻⁶⁶. Thus, our data on the high rate of social (horizontal) transmission in the *Molothrus* system potentially

represent a more general pattern in feather mites that needs to be further investigated in other bird systems as well.”

R3.Q3. I have uploaded an annotated version of the manuscript pdf that includes my comments as pop-up boxes and text edits. Below I list the more important of them in order of their occurrence in the ms; however, the authors should read through the annotated pdf to see all of my suggestions for changes to syntax, word choice, grammar, spelling and punctuation.

R3.A3. We thank all the corrections the reviewer suggested. We have adopted all the grammatical changes.

R3.Q4. Abstract and throughout: depending on journal formatting style, you may need to write out genus names in full at beginning of sentences rather than using abbreviations.

R3.A4. To our knowledge, there are no rules regarding the abbreviations of the genus name in scientific species names. We however added full names at the beginning of sentences.

R3.Q5. Introduction, line 45: The highlighted part of this long sentence is rather an awkward construction. I suggest giving the highlighted section its own sentence, in which you elaborate more about what Refs 12-14 are referring to.

R3.A5. Text was split as the reviewer suggested (other modifications were done as per R3.Q5).

Old text: “Generally, disagreement between host and symbiont phylogenetic branching patterns at the macroevolutionary scale may be caused by several evolutionary events, such as duplication (speciation of a symbiont within a single host species), sorting (extinction and missing the boat), failure to speciate, and host switching (or host shift), with the last one being generally a common event^{1,5,10,11}, but see¹²⁻¹⁴.”

New text: “Generally, cophylogenetic incongruence (i.e., the disagreement between host and symbiont phylogenetic branching patterns at the macroevolutionary scale) may be caused by several evolutionary events, such as duplication (speciation of a symbiont within a single host species), sorting (extinction and missing the boat), failure of the symbiont to speciate, and host switching (or host shift)¹⁶. Among these events, host switching is typically a common event^{1,8,13,16,17}.”

R3.Q6. Introduction, line 59: This is not true for Epidermoptidae such as *Myialges* spp. that are specialized for dispersal via hippoboscid flies. This is mentioned later but perhaps you shouldn't imply at this point that no feather mites are adapted for long-range horizontal transmission.

R3.A6. Changed as suggested.

Old text. “These symbiotic organisms spend their entire life cycle on the host (full-time, obligate symbionts) and, not only lack a specialized dispersal stage but also seem to lack any other adaptations for long-range dispersal between hosts^{28,30,31}.”

New text: “These symbiotic organisms spend their entire life cycle on their host (full-time, obligate symbionts). With a few exceptions (e.g., some skin mites), they do not have a specialized dispersal stage and seem to lack any other adaptations for long-range dispersal between hosts^{33,35,36}.”

R3.Q7. Introduction, lines 65-71: A long run-on sentence! Please break into two sentences for ease of reading.

R3.A7. We re-wrote and split this sentence as follows:

Old text: “Given the discordance of current biologically informed expectations (i.e., vertical transmission should be common, hence host and symbiont phylogenies should be congruent vs. observations of widespread phylogenetic incongruence while mites maintain high host specificity), it is important to understand the relative contribution of two types of conspecific transmission: vertical and horizontal (q_{vc} and q_{hc}), promoting cophylogenetic concordance and high host specificity vs interspecific transmission (q_{vi} and q_{hi}), that can generate cophylogenetic discordance and low host specificity (Fig. 1).”

New text: “Thus, the current biological expectations are in conflict: one suggests that vertical transmission should be prevalent, leading to congruence between host and symbiont phylogenies, while observations show widespread phylogenetic incongruence among mites, despite their high host specificity. Therefore, it is crucial to understand the relative contribution of two types of conspecific transmission: vertical and horizontal (q_{vc} and q_{hc}), promoting cophylogenetic concordance and high host specificity vs interspecific transmission (q_{vi} and q_{hi}), that can generate cophylogenetic discordance and low host specificity (Fig. 1).”

R3.Q8. Introduction, lines 72-73: Maybe vertical transmission can be accurately measured, but such measurements are rare. Please cite publications that include assessing transmission of feather mites from parents to offspring.

R3.A8. We added two references [25, 39]:

“In feather mites, vertical conspecific transmission (q_{vc}) occurring from parents to chicks during the nesting period can be accurately measured^{25,39}.”

[25] Doña, J. *et al.* Vertical transmission in feather mites: insights into its adaptive value. *Ecol. Entomol.* **42**, 492–499 (2017).

[39] Mironov, S. V. & Malyshev, L. L. Dynamics of infection the Chaffinch nestlings *Fringilla coelebs* with feather mites (Acari: Analgoidea). *Parazitologiya* **36**, 356-374. [In Russian with English summary] (2002).

R3.Q9. Introduction, ~ line 78: Please briefly provide some biological information about *M. bonariensis*, e.g., its geographical distribution, what families of birds it parasitizes, etc.

R3.A9. We included the following sentence with biological information about *M. bonariensis* in the Introduction:

“This is a generalist bird parasitizing more than 90 different passerine species in 17 families in South America and beyond⁴¹⁻⁴³”

R3.Q10. Introduction, line 78 and elsewhere in ms: Depending on journal formatting guidelines, you may need to provide taxonomic authorities when giving scientific names of species.

R3.A10. The formatting guideline does not require citing the name of taxonomic authorities.

R3.Q11. Introduction, line 109: Here cite some of the literature that makes these assumptions of dominance of horizontal transmission.

R3.A11. This is our assumption based on our hypothesis (H_i). However, we do provide references for published works that assume a greater role of vertical transmission. The sentence was modified as follows (highlighted text inserted):

Old text: “Hypothesis H1 favors a greater role of horizontal transmission than it is assumed in the literature (H1: case 2).”

New text: “Hypothesis H_i (Fig. 1B: H_i) favors a greater role of horizontal transmission in maintaining host specificity than it is assumed in the literature, e.g.,^{25,39}.”

R3.Q12. References: see annotated pdf for corrections to some references.

R3.A12. Changed as requested.

R3.Q13. Results, line 121: Reference 31 indicates that the sister of *M. bonariensis* is not *M. ater*, but rather the small clade of (*M. aeneus*+*M. ater*).

R3.A13. The correct reference was inserted [Jetz et al., 2012]. Ref 31 was based on the phylogeny of Powell et al. (2014), which was not resolved, e.g. support for the “clade” *M. aeneus*+*M. ater* was 50% and there was no nuclear DNA data for *M. bonariensis* and *M. aeneus*.

Jetz, W., Thomas, G. H., Joy, J. B., Hartmann, K., & Mooers, A. O. (2012). The global diversity of birds in space and time. *Nature*, 491(7424), 444-448. doi:10.1038/Nature11631

Powell, A. F. L. A., Barker, F. K., Lanyon, S. M., Burns, K. J., Klicka, J., & Lovette, I. J. (2014). A comprehensive species-level molecular phylogeny of the New World blackbirds (Icteridae). *Molecular Phylogenetics and Evolution*, 71, 94-112.

R3.Q14. Results, line 130: "Quill-and-skin mites (QSM) – this classification is based mostly on the literature: these mites are naturally rare (and have low prevalence/abundance)...". Provide references for this statement. Also, do you think this is **truly** low prevalence and abundance for skin and quill-dwelling analgoids and pterolichoids? Or just **apparent** rarity due to feather-ruffling methods being a poor way to recover these mites?

R3.A14. These mites have truly low prevalence and abundance based on our experience (visual inspection of birds for parasites) and literature data. References for this statement are provided (see below). We also better defined this group.

Old text="Quill-and-skin mites (QSM) – this classification is based mostly on the literature: these mites are naturally rare (and have low prevalence/abundance), live on the host skin (e.g., *Microlichus*, *Metamicrolichus*, *Strelkoviacarus*) and inside quills (*Dermoglyphus*). In contrast, mites in the two former categories inhabit vanes or down feathers. The skin- inhabiting mites have the ability to be horizontally transferred across interspecific and intraspecific hosts via phoresy on louse flies (Diptera: Hippoboscidae)32–34."

New text="Quill-and-skin mites (QSM) – inhabit the host skin (e.g., *Microlichus*, *Metamicrolichus*, *Strelkoviacarus*) or live inside feather quills (*Dermoglyphus*). **These mites are**

grouped together because of their restricted contact transmission in comparison to that of plumage feather mites (groups i and ii). For example, when two bird individuals come into contact, the likelihood of sharing skin or quill mites is significantly lower as compared to that of plumage mites. Furthermore, skin mites are naturally rare and have low prevalence and abundance⁷³ in comparison with plumage mites⁷⁴. Some skin-inhabiting mites also have the ability to be horizontally transferred across interspecific and intraspecific hosts via phoresy on louse flies (Diptera: Hippoboscidae)^{36,78,79}. As a result, in terms of their transmission patterns, these mites cannot be directly compared to plumage-inhabiting feather mites (groups i and ii). “

[73] Hill, D. S., Wilson, N., & Corbet, G. B. (1967). Mites associated with British species of *Ornithomya* (Diptera: Hippoboscidae). *Journal of Medical Entomology*, 4(2), 102-122. doi:10.1093/jmedent/4.2.102

[74] Matthews, A. E., Larkin, J. L., Raybuck, D. W., Slevin, M. C., Stoleson, S. H., & Boves, T. J. (2018). Feather mite abundance varies but symbiotic nature of mite-host relationship does not differ between two ecologically dissimilar warblers. *Ecology and Evolution*, 8(2), 1227-1238. doi:10.1002/ece3.3738

R3.Q15. Results, line 135: I agree that it's a good idea to separately consider the skin mites that have been documented to transfer horizontally via hippoboscids; however, there is no clear justification for including quill mites together with the skin mites in this category. What is your rationale?

R3.A15. Skin mites and quill mites are grouped together due to their limited transmission through direct contact (this is in contrast to plumage feather mites which can easily disperse through contract transmission). To illustrate this, when two bird individuals come into contact, the likelihood of sharing skin or quill mites is significantly lower as compared to that of plumage mites (plumages may be touching but skin or inner contents of quills are not). See also R3.A14.

R3.Q16. Results, line 220: Why suggest social contact rather than just repeated transmission from foster parents to *M. ater* chicks? Or do you mean that *P. egglestoni* might be transferred between *M. ater* due to social contacts? Either way, it is ambiguous as written.

R3.A16. Here we suggested that these mites can be transmitted by both vertical and horizontal routes, we clarify this sentence to remove the ambiguity:

Old text: “*Proctophyllodes egglestoni*, on the other hand, experiences ongoing mite transmission from foster parents to *M. ater*, possibly through social contact (Fig. 4D).”

New text: “*Proctophyllodes egglestoni*, on the other hand, has experienced ongoing transmission from foster parents to *M. ater*, possibly through both vertical and horizontal routes as some foster parent species may also form mixed flocks with *M. ater*” (Fig. 3C)”

R3.Q17. Discussion, lines 257-259: "In the cowbird system, all *Molothrus*-specific mites were transmitted horizontally via conspecific contacts, which likely occurred gradually with host age as the birds experience an increasing rate of social contact. Feather mites associated with the Australian bushturkeys (Galliformes: Megapodiidae), a host that has minimal parent-to-offspring contact (eggs are buried, young birds are fully fledged on hatching) do display this pattern as

well." By saying 'as well', you imply that you have demonstrated gradual accumulation of mite taxa as the *M. bonariensis* aged. But I don't see such data in this manuscript.

R3.A17 This sentence was re-written to remove the ambiguity as follows :

“In the cowbird system, all *Molothrus*-specific mites were transmitted horizontally via conspecific contacts, which likely occurred gradually with host age as the birds experienced an increasing rate of social contact. This has been shown in feather mites associated with the Australian bushturkeys (Galliformes: Megapodiidae), a host that has minimal parent-to-offspring contact (eggs are buried, young birds are fully fledged on hatching)⁵⁸”

R3.Q18. Discussion, line 270: Provide a suggestion of how this cowbird-to-foster transfer might occur (e.g., by fighting between female cowbirds and blackbirds when the cowbirds try to lay eggs in host nests?).

R3.A18. Suggestion added (see highlighted text):

“For example, an exceptional multihost generalist mite, *Proctophyllodes egglestoni*, can probably be transferred from *M. ater* to its forest parents (*Agelaius phoeniceus*) and back through aggressive (e.g., nest defense) and gregarious (e.g., mixed-flocks) behaviors⁵⁹. These ongoing transmissions are supported by the shallow genetic distances between the mites from the two hosts, COX1 K2P = 0.2% (Fig. 3D).”

R3.Q19. Discussion, lines 274-287: This paragraph seems rather poorly organized and internally repetitive. Please organize more logically and concisely.

R3.A19. We reorganized and simplified this paragraph as follows:

Old text: “The literature also suggests the presence of horizontal conspecific transmission of symbionts in other brood parasitic birds, albeit with no quantitative data, or phylogenetic analyses. Two other brood-parasitic lineages, European cuckoos (*Cuculus canoris*) and indigobirds (*Vidua*), have their own symbionts (mites or lice)⁴⁷⁻⁵¹, suggesting that they also can be acquired via horizontal transmission from conspecifics. Horizontal transmission has also been observed as the main transmission route even in non-brood parasitic systems, such as in feather mites from the red-billed choughs⁵², barn swallows⁵³, and in feather lice from bee-eaters⁵⁴. A mixture of horizontal and vertical transmission (parental or foster parental care), as exemplified by the presence of specific and foster parent symbiotic arthropods, has been recorded in cowbirds and other systems, but again without quantitative data^{45,47,55-57}. Some studies on birds with parental care also indicated that horizontal transfers (via host interspecific contact) in feather mites may be more frequent than previously thought^{16,19,58}. Thus, our data on the high rate of social transmission in the *Molothrus* system potentially represent a more general pattern that needs to be further investigated in other bird systems as well.

New text: “Previous studies have suggested the presence of horizontal conspecific and vertical interspecific transmission in cowbirds and other brood parasitic birds by the presence of host-specific or parent-specific symbionts, albeit with no quantitative data or phylogenetic analyses^{57,60-63}. Even in non-brood parasitic systems, horizontal conspecific transmission has also been observed as the main transmission route of symbionts, such as feather mites from red-billed choughs⁶⁴, barn swallows⁶⁵, and in feather lice from bee-eaters⁶⁴⁻⁶⁶. Thus, our data on the high rate of social (horizontal) transmission in the *Molothrus* system potentially represent a more general pattern in feather mites that needs to be further investigated in other bird systems as well.”

R3.Q20. Discussion, lines 317-318: Please cite some of the references in which this 'traditional view' of the 'sole force' is explicitly stated.

R3.A20. We replaced 'sole force' with 'main force' and provided references:

Old text: “Our data provide further evidence challenging the traditional view of the importance of vertical transmission as the sole force generating host specificity and phylogenetic congruence in feather mite systems.”

New text: “Our data therefore provide evidence challenging the traditional view of the importance of vertical transmission as the main force generating host specificity on a microevolutionary scale⁷¹.”

[71] Fisher, R. M., Henry, L. M., Cornwallis, C. K., Kiers, E. T. & West, S. A. The evolution of host-symbiont dependence. *Nat. Commun.* **8**, 1–8 (2017).

R3.Q21. References: see annotated pdf for corrections to some references.

R3.A21. Changed as requested.

R3.Q22. Table 1: Not clear what 'prevalence by mite record' means. Is it 'proportion of all mite specimens'?

R3.A22. Yes, it is the proportion out of the total mite records (n=365), we changed the previous headings “Prevalence by mite record” and “Host-parasite records” to “Proportion of mite records based on p/a (%)” and “Mite p/a records”

p/a = presence/absence

The headings “Prevalence by mite record” -> “Proportion of mite records, p/a (%)”

“Host-parasite records” ->“Mite records, p/a”

p/a = presence/absence

R3.Q23. Table 2: Most (or maybe all) of the zeros in this table should be 'not applicable' (n/a) instead, e.g., it is impossible for 'alien only' mites to have any representation in the 'Molothrus-specific' column.

R3.A23. Changed as requested.

Figure 1. (A) Conspecific (vertical/horizontal) vs interspecific (vertical/horizontal) transmissions in different bird systems (brood parasites and non-brood parasites); (B) two hypotheses on how different symbiont transmission types may affect the mite host specificity and cophylogenetic congruence with their avian hosts. Our null hypothesis (H₀) proposes that a lower conspecific transmission rates (in comparison to the interspecific rate) results in lower host specificity, lower number of codivergence events, and higher number of host switches; whereas a higher rate of conspecific transmission (H₁) is expected to result in higher host specificity, higher number of codivergence events, and lower number of host switches.

Figure 2. Maximum Likelihood phylogeny of feather mites based on 6 genes (5 nuclear, 1 mitochondrial). Nodal support higher than 75% (estimated by ultrafast bootstrap with 1000 replicates) is shown by thicker lines. The major mite lineages, Trouessartiidae, Proctophyllodinae and Pterodectinae, are highlighted. Mites from cowbirds (*Molothrus bonariensis* and *M. ater*) and their foster parent birds: finches (*Sicalis flaveola* and *S. luteola*) and other Icteridae are highlighted. Portions of this phylogeny, exemplifying important host shifts to *Molothrus* are given for each mite lineage on insets (A-D); these cases are further considered in detail in Fig. 3.

Figure 3. A–D Mite molecular phylogenetic trees, cophylogenetic reconciliation analysis and schematic explanations of cophylogenetic scenarios (for complete phylogenies, see Fig. 2; and Supplementary Figs. 1-5). Nodal support is given for each branch (ultrafast bootstrap, 1000 replicates). Molecular CO1 K2P genetic distances are given as percentages. See the Result section for detailed explanation of cophylogenetic scenarios.

REVIEWERS' COMMENTS:

Reviewer #1 (Remarks to the Author):

All my minor comments from the previous version have been successfully addressed. I find this study very interesting and provide novel insights into the importance of conspecific horizontal transmission on host-symbiont specificity.

Reviewer #2 (Remarks to the Author):

The authors addressed the concerns raised properly. I do not have anything else to add.

Reviewer #3 (Remarks to the Author):

The authors have done an excellent job of revising their manuscript to improve clarity, both in the text and in tables and figures. I have only a couple of comments and in addition have made a few edits on the uploaded annotated manuscript. The comments are:

line 367: As worded it isn't clear whether you mean (1) just a subset of macroevolutionary patterns, in which case you should remove the comma after 'patterns' and use 'that' rather than 'which', or (2) **all** macroevolutionary patterns, in which case you need to add a comma after 'events'. (1) "...macroevolutionary patterns **that** are based on rare coevolutionary events cannot be easily generalized..." or (2) "...macroevolutionary patterns, which are based on rare coevolutionary events, cannot be easily generalized...".

line 398: *Molothrus* spp. are not included in any of the main trees in this paper (Jetz et al. 2012). Are the relationships among the *Molothrus* species indicated in one of the Supplementary files for this paper (of which there seem to be at least 100 files)? If so, please explicitly state where this sister relationship is shown in Jetz et al. (2012).

Table 2, last row in 'host individuals' column: why doesn't this equal 143? Only 1 of the 144 cowbirds had only QSM mites, whose mode of transmission was undetermined. Or maybe this means that some birds had no mites at all? If I'm not understanding what this value is supposed to indicate, it is likely to be unclear to other readers as well, and so it should be better explained in the table caption or as a footnote.

Here are our responses to the Reviewer 3 final comments (see below). We do appreciate the reviewers' input and their effort to improve the manuscript.

Thank you for your continued support in the reviewing process.

Sincerely,

R3.Q1. Line 358 'invaders' seems both too strong and too pejorative a word. I suggest 'colonists' instead, or at least put the word 'invaders' in single quotes to indicate that you are meaning this figuratively.

R3.A1. As suggested, we changed "invaders" to "colonists":

New text: "These symbionts persist on *M. bonariensis* at high prevalence and abundance despite the constant influx of new diverse mite **colonists** transmitted vertically from over 90 species of foster parents."

R3.Q2. As worded it isn't clear whether you mean (1) just a subset of macroevolutionary patterns, in which case you should use 'that' rather than 'which', or (2) all macroevolutionary patterns, in which case you need to add a comma after 'events'. (1) "...macroevolutionary patterns that are based on rare coevolutionary events cannot be easily generalized..." or (2) "...macroevolutionary patterns, which are based on rare coevolutionary events, cannot be easily generalized...".

R3.A2. This was an appositive clause that provides additional information about the noun ("patterns") in the sentence. As such, it should be set off by commas to show that it is supplementary to the sentence's main structure. We, therefore, inserted a comma after the word "events" as suggested by the reviewer in (2):

New sentence: "This suggests that macroevolutionary patterns, which are based on rare coevolutionary events, cannot be easily generalized from short-term evolutionary trends, such as transmission mode and rates."

R3.Q3. *Molothrus* spp. are not included in any of the main trees in this paper (Jetz et al. 2012).. Are the relationships among the *Molothrus* species indicated in one of the Supplementary files for this paper (of which there seem to be at least 100 files)? If so, please explicitly state where this sister relationship is shown in Jetz et al. (2012).

R3.A3. Sister-group relationship of *Molothrus bonariensis* and *M. ater* are shown in this supplementary file of Jetz et al. (2012): P21.MCC.tre (posterior probability=1.00).

We included this information in our Methods section text as follows:

Old text: "specimen of *Molothrus ater*, which is sister to *M. bonariensis* ..."

New text: "... specimen of *Molothrus ater*, which is sister to *M. bonariensis* (see supplementary file P21.MCC.tre in ref ⁷²) ...

⁷²Jetz, W., Thomas, G.H., Joy, J.B., Hartmann, K., Mooers, A.O., 2012. The global diversity of birds in space and time. *Nature* 491, 444-448.

R3.Q4. Why doesn't this equal 143? Only 1 of the 144 cowbirds had only QSM mites, whose mode of transmission was undetermined. Or maybe this means that some birds had no mites at all? If I'm not correctly understanding what this value is supposed to indicate, it is likely to be unclear to other readers as well, and so it should be better explained in the table caption or as a footnote.

R3.A4. 139 is the sum of all bird individuals with mite records. Five birds out of the total 144 analyzed birds had no mites, totaling 139 birds with mite records. We included this information in the Table's heading to avoid misunderstandings, as follows:

New text: Data are summarized from 365 mite records (=total of transmission cases) sampled from 144 bird individuals. Of them, 139 bird individuals had mites, while 5 bird individuals lacked any mites.

Old text: Data are summarized from 365 mite records (=total of transmission cases) sampled from 144 bird individuals.